# A convolutional neural network STIFMap reveals associations between stromal stiffness and EMT in breast cancer

Connor Stashko[1,2], Mary-Kate Hayward [1,2,12], Jason J. Northey [1,2,12], Neil Pearson[3], Alastair J. Ironside[4], Johnathon N. Lakins[1,2], Roger Oria [1,2], Marie-Anne Goyette[5], Lakyn Mayo [6], Hege G. Russnes [7,8], E. Shelley Hwang[9], Matthew L. Kutys [6,10], Kornelia Polyak [5] & Valerie M. Weaver [1,2,10,11] ✉

Intratumor heterogeneity associates with poor patient outcome. Stromal stiffening also accompanies cancer. Whether cancers demonstrate stiffness heterogeneity, and if this is linked to tumor cell heterogeneity remains unclear. We developed a method to measure the stiffness heterogeneity in human breast tumors that quantifies the stromal stiffness each cell experiences and permits visual registration with biomarkers of tumor progression. We present Spatially Transformed Inferential Force Map (STIFMap) which exploits computer vision to precisely automate atomic force microscopy (AFM) indentation combined with a trained convolutional neural network to predict stromal elasticity with micron-resolution using collagen morphological features and ground truth AFM data. We registered high-elasticity regions within human breast tumors colocalizing with markers of mechanical activation and an epithelial-to-mesenchymal transition (EMT). The findings highlight the utility of STIFMap to assess mechanical heterogeneity of human tumors across length scales from single cells to whole tissues and implicates stromal stiffness in tumor cell heterogeneity.

Intratumor heterogeneity (ITH) is a feature of tumors including breast cancers[1–4]. Tumor heterogeneity predicts poor patient outcome as diversification of genetic, phenotypic and behavioral characteristics within a tumor support progression, metastasis, and treatment resistance[5–7]. Accordingly, much effort has been directed towards defining ITH and clarifying how it drives tumorigenesis[8,9]. Towards this goal, the ability to decipher the causal relationship between cancer heterogeneity and tumor phenotype relies heavily on the availability of accurate and quantitative methods with which to measure and analyze individual features of the tumor.

Tumor tissue variability is mediated, in part, by intrinsic stochastic gene expression as well as by genetic and epigenetic differences in the transformed cells. Approaches including genetic tags and high-throughput sequencing have permitted researchers to detect genomic abnormalities at the single cell level to provide important insights into clonal evolution and have linked these findings to patient

[1]Department of Surgery, University of California, San Francisco, CA, USA. [2]Center for Bioengineering and Tissue Regeneration, University of California, San Francisco, San Francisco, CA, USA. [3]Harvey Mudd College, Claremont, CA, USA. [4]Department of Pathology, Western General Hospital, NHS Lothian, Edinburgh, UK. [5]Department of Medical Oncology, Dana-Farber Cancer Institute, Boston, MA, USA. [6]Department of Cell and Tissue Biology, School of Dentistry, University of California, San Francisco, San Francisco, CA, USA. [7]Department of Pathology and Department of Cancer Genetics, Institute for Cancer Research, Oslo University Hospital, Oslo, Norway. [8]Institute for Clinical Medicine, University of Oslo, Oslo, Norway. [9]Department of Surgery, Duke University Medical Center, Durham, NC, USA. [10]UCSF Helen Diller Comprehensive Cancer Center, University of California, San Francisco, San Francisco, CA, USA. [11]Department of Radiation Oncology, Eli and Edythe Broad Center of Regeneration Medicine and Stem Cell Research, University of California, San Francisco, San Francisco, CA, USA. [12]These authors contributed equally: Mary-Kate Hayward, Jason J. Northey. ✉e-mail: valerie.weaver@ucsf.edu

survival[10–12]. Spatial RNA sequencing (RNAseq) analysis has revealed underlying spatial associations between stress response gene expression profiles in cancer cells and inflammatory fibroblast gene signatures[13]. Indeed, tumors are organs comprised of transformed cells interacting with a diverse cellular and acellular stroma. Consistently, in situ multiplexing approaches have revealed broad diversity with respect to the frequency and phenotype of tumor-infiltrating immune cells and have used these findings to predict patient outcomes and immune checkpoint inhibitor responsiveness[14]. In situ immunofluorescence has also illustrated wide variability in oncogenic signaling, cellular metabolism and stress responsiveness between the epithelial and stromal cells within the hypoxic tumor core, and those cells that localize to the fibrotic tumor periphery, that predict treatment response in patients[15,16].

A feature of all solid tumors is the remodeled and crosslinked extracellular matrix (ECM) that generates a stiffened, fibrotic stroma characterized by markedly reorganized interstitial collagens[17]. A stiff ECM modifies cell and nuclear shape, disrupts tissue organization, promotes cell growth, viability, and invasion, alters gene expression, and can induce an epithelial-to-mesenchymal transition (EMT) in cells cultured in two- and three-dimensional substrates[18]. Within experimental tumors in vivo, the stiffened tumor ECM promotes solid stress, disrupts vascular integrity to drive hypoxia and tumor aggression, and compromises drug delivery[19]. The stiff in vivo tumor ECM also increases cytokine and chemokine expression to promote myeloid cell infiltration and may impede CD8 T cell infiltration, migration, and function[20]. Clinically, the level of tissue fibrosis correlates with worse patient outcome, and in situ analysis of human breast cancer tissues revealed that a stiff, fibrotic ECM associates with tumor progression as well as clinical subtype[21–23]. Whether stromal stiffness heterogeneity tracks with tumor cell heterogeneity and contributes to human breast cancer aggression remains unclear.

To clarify links between stromal stiffness heterogeneity and tumor cell heterogeneity, approaches are needed that can be combined with state-of-the-art spatial genomics, proteomics, and multiplexing protocols[24–26]. Although techniques do exist with which to monitor ECM heterogeneity and organization including H&E, second-harmonic generation (SHG), trichrome, and picrosirius red (PS red) staining, none can be directly combined with immunostaining on the same slide[21,27,28]. Moreover, these protocols do not provide quantitative information regarding mechanically soft and stiff regions within the tumor. Methods to directly measure ECM stiffness include shear rheology, magnetic resonance elastography, sonoelastography, and unconfined compression analysis[29–32]. However, these approaches do not provide high resolution spatial and morphological information, particularly with respect to the state of the collagenous ECM. Although stromal stiffness can be measured directly with sub-micron resolution using AFM, current AFM methods are time-consuming, poorly resolved spatially, and require specialized equipment not readily available to most research and clinical laboratories[33,34]. An automated AFM developed by Plodinec and colleagues can rapidly quantify the material properties of tumor biopsies, but the method does not provide imaging of the probed tissue, nor micron resolution positioning of where precisely measurements were taken. Thus, while the approach is useful for characterizing cell and tissue biomechanical properties, it is not feasible to link the AFM elasticity measurements obtained of the ECM with biological markers of tumor and stromal phenotype, genotype, and heterogeneity[35]. Finally, all of the current approaches used to quantify cell and stromal stiffness require manipulation of either fresh or cryopreserved tissue, precluding comprehensive spatial analysis of elasticity in archived formalin-fixed paraffin-embedded (FFPE) sections in tissue banks.

Here, we present an approach termed Spatially Transformed Inferential Force Map (STIFMap), that is able to visualize the heterogeneous stiffness landscape of normal and tumor breast tissues with micron-resolution and can also spatially register this tension phenotype together with biological markers of tumor progression and histophenotype at micron resolution. The method works on both cryopreserved and FFPE tissues and employs a single quick, inexpensive collagen stain that is visualized with standard fluorescence microscopy. The approach permits simultaneous quantification of the tension landscape of the stromal ECM together with co-staining for cell or ECM biomarkers of interest, and lends itself to quick assessment of the impact of biophysical ECM heterogeneity on tumor progression. The method can be readily integrated with spatial proteomics and genomics, as well as standard protein marker multiplexing protocols. To illustrate the potential of the approach, we applied STIFMap to explore the relationship between stromal stiffness heterogeneity and markers of mechanical activation and tumor progression in human breast cancers. We were able to link tissue mechanics with indicators of mechanosignaling and biomarkers of EMT previously implicated in tumor progression and treatment resistance[36,37]. The results highlight the potential utility of using stromal biophysical features to monitor tumor heterogeneity and clarify links to tumor behavior and possibly even patient outcome.

## Results

### Design and development of an automated AFM system

AFM has emerged as the method of choice to spatially analyze stromal stiffness at micron resolution in tumors[38,39]. However, executing AFM analysis is cumbersome, specialized, and not easily amenable to spatial registration with sequential in situ analysis and imaging. To improve upon these pitfalls, we developed AutoAFM to facilitate high-throughput, spatially-resolved acquisition of AFM data. We automated AFM movements by affixing servo motors onto the X and Y translation knobs of the AFM stage with custom-made, 3D-printed motor mounts (Fig. 1a, b, Supplementary Table 1, Supplementary Fig. 1a, b, Methods). Scripts were developed to enable the AFM to move along a user-specified path (Fig. 1c, d). The system was designed so that as the AFM moves from one point to the next, a feedback loop reports on the current position of the AFM and fine-tunes movements to poke the specimen within a user-designated tolerance of the desired positions. All movements and imaging were designed to be conducted using epifluorescence of propidium iodide-stained (PI) cells to guide the measurements. This strategy was chosen to remove artifacts from the cantilever shadow that could potentially be introduced into the images during stitching (Supplementary Fig. 1c). The system was engineered so that a completed AutoAFM scan will provide the location of each AFM force curve acquired over the tissue section being measured (Fig. 1e, f). The AutoAFM was designed such that scans can be acquired across as many points as the operator desires and are only spatially limited by the overall X-Y range of the AFM stage.

### Assessment of AutoAFM precision and validation of AFM measurements

To validate movements of the AutoAFM, a series of elevated PDMS beams of varying width were fabricated using photolithography followed by PDMS soft lithography (Supplementary Fig. 1d). Using this strategy, the height at which the AFM contacts the sample is known, so force curves collected on the beams registered as much higher than those collected on the surrounding PDMS surface. To determine the resolution limit of AutoAFM, we used the AFM to 'walk' along each beam and measured the accuracy of the AFM to contact the beam at each width. The measurements indicated that movements of the automated AFM are precise to within a few microns (Supplementary Fig. 1e).

The Young's Modulus of an AFM cantilever is calibrated before measurements are performed (Methods). Nevertheless, the cantilever modulus can change over the course of data collection due to protein and cell debris deposition onto the cantilever. To ensure that the

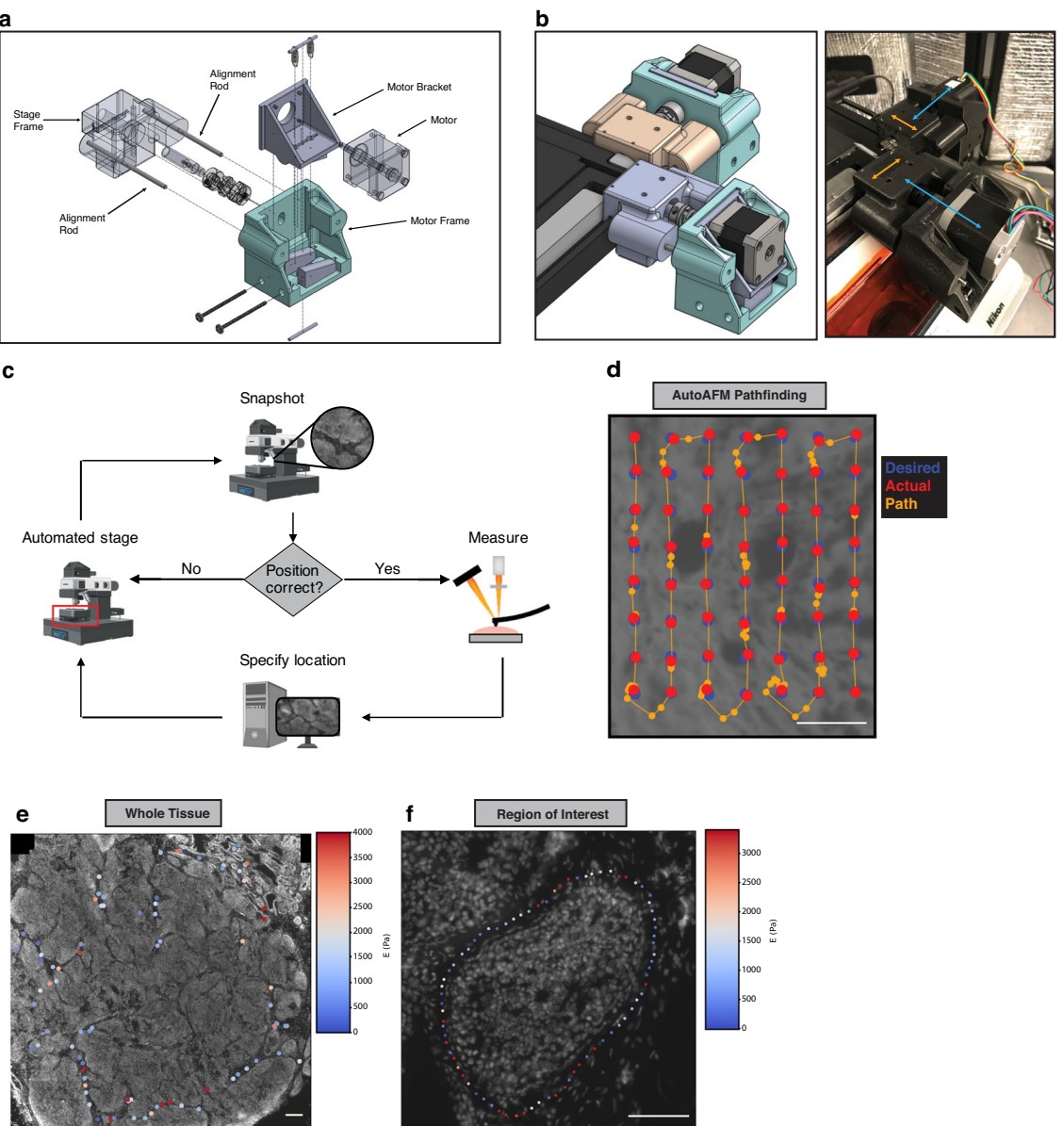

**Fig. 1 | Overview of automated AFM acquisition system. a** Technical drawing of motor mount for interfacing servo motors with AFM translation knobs. **b** Assembled motor mounts. The Stage Frame slides along the edge of the stage (orange arrows) while the Motor Frame slides along the Alignment Rods, towards and away from the Stage Frame as the knobs turn (blue arrows). **c** Schematic of AutoAFM feedback system. **d** Example of AutoAFM feedback with desired AFM sampling positions (blue), actual AFM positions (red), and AFM path of movement and positions outside of the desired positions (orange). Representative images of AutoAFM collecting AFM measurements over a whole tissue (**e**) and a region of interest (**f**) in a breast tumor section. 200 experiments were conducted obtaining reproducible results. Scale bar, 100 μm.

stiffness of the cantilever remained consistent throughout the measurements, we measured the elasticity of polyacrylamide (PA) gels of known Young's Moduli before and after probing each sample. By comparing the stiffnesses pre- and post-sampling, we observed a 1:1 relationship (Supplementary Fig. 2a), indicating no changes in stiffness of the cantilever had occurred during the measurements. If the tip was contaminated, the data would have been consistently above/below the 1:1 line (140 Pa gels $p$-value = 0.551 and 1 kPa gels $p$-value = 0.970. Wilcoxon Rank-Sum Test). Additionally, the elasticities of the PA gels used were also validated independently using shear rheology (Supplementary Fig. 2a). AFM measurements on each tissue were collected over 1–2 h. To verify that tissues did not degrade over the timespan of AFM measurements, we collected force curves at the same tissue positions over a defined length of time. AFM measurements collected in the same positions every 30 min for three hours revealed no noticeable

differences in elasticity, indicating that tissue degradation did not occur over the timespan that AutoAFM measurements were acquired for the measurements conducted in the current study (Supplementary Fig. 2b). Because tissues are viscoelastic substrates, the rate of force loading impacts the resulting force curve[40]. Accordingly, highly viscous substrates will appear stiffer when poked faster if they are assumed to be purely elastic. To address this potential anomaly, an AFM velocity of 2 μm/s was chosen since Young's Moduli measurements are constant at this rate (Supplementary Fig. 2c).

To avoid any potential for tip fouling, we chose AFM cantilevers that were triangular with 5 μm spherical beads incorporated onto the cantilever tip (Supplementary Fig. 2d[41];). To stitch together images taken during AutoAFM, the bead location was estimated for each image. To estimate the bead location within the image, the average image for each AutoAFM scan was taken, which revealed a faint but

distinct outline of the cantilever (Supplementary Fig. 2e). This occurred because the stronger PI signals from the cells move during Auto-AFM acquisition, but the faint cantilever image remains in the same position throughout. Five cantilevers with known bead locations were aligned with the average scan images to indicate the actual position of the bead during imaging (Supplementary Fig. 2f, g).

## Overlaying autoAFM measurements onto confocal images

In an effort to ensure visual spatial alignment between tissue morphological features and elasticity measurements, the AutoAFM measurements were overlaid with nuclear staining via alignment with AutoAFM PI positions. DAPI measurements were collected via confocal imaging at either 40× or 63× magnification, while AutoAFM PI images were collected at 20× magnification. A pipeline was then developed to translate low-resolution AutoAFM images onto high-resolution DAPI imaging (code available on GitHub). To do so, the two images were first manually pre-aligned so that the fields of view were similar, and the confocal DAPI image was down sampled to more closely resemble the resolution of the AutoAFM image (Supplementary Fig. 3a). Thereafter, a Fourier-Mellin Transform was applied to determine the scale and rotation of the AutoAFM image relative to the DAPI image[42]. Finally, translation between the two images was computed using phase contrast cross-correlation. Using this transformation matrix, AFM positions were mapped onto the high-resolution images (Supplementary Fig. 3b). The average mapping error was found to be 2.57 μm, estimated by monitoring nuclei positions before and after transformation (95% confidence interval: 2.09–3.06 μm) (Supplementary Fig. 3c).

## Deep learning model of tissue Young's Modulus from collagen morphology

Interstitial fibrillar collagens are the major structural component of breast tissue[43–46]. As such, we reasoned that the elasticity of breast tissue could be inferred based on the morphology of interstitial collagen fibers, particularly given that stiff collagen fibers are thick and highly linear whereas more compliant collagen fibers are typically more dispersed, relaxed, and visually present as wavy fibers[47]. Although most investigators have used SHG imaging or PS red staining to visualize interstitial fibrillar collagens, collagen SHG imaging is susceptible to interference from additional fluorophores on co-stained tissues[48], and PS red coloring depends on the angle of the slide on the microscope relative to the polarizer[28]. Accordingly, we chose to stain the collagens using the collagen-binding adhesion protein 35-Orange Green 488 fusion protein (CNA35-OG488; CNA)[49, 50]. CNA contains two subdomains, N1 and N2, which engage in a 'collagen hug' around triple helical collagen in the ECM[51]. CNA staining was selected because it is cheap to produce, plasmid sequences are freely available online, and the stain is not species specific[52]. The CNA stain is also rapid and easy to perform, and can be viewed on conventional fluorescent microscopes, lending itself to standard research laboratories as well as a clinical lab format.

To register collagen morphological features with tissue stiffness, a convolutional neural network (CNN) was applied using CNA and DAPI imaging as the inputs and the corresponding AFM measurements as the output (Fig. 2a). A CNN was chosen due to its superiority compared to alternative models with image classification tasks[53]. We reasoned that a CNN would be able to learn how factors such as collagen fiber linearity, thickness, and proximity to cells impacts elasticity better than alternative models. Different CNN architectures were applied to predict tissue elasticity including ResNet, DenseNet, and models discovered using Neural Architecture Search, but the best performance came from an AlexNet modified for a regression output instead of classification (https://github.com/pytorch/vision/blob/main/torchvision/models/alexnet.py).

Neural networks are data hungry, such that their performance is greatly improved when more data is utilized. When given a small amount of training data, neural networks tend to over-fit the training dataset and emphasize features that do not generalize well. To address this, we artificially enlarged our training dataset of a thousand data points by applying random rotations, mirroring, and adjustments to brightness, contrast, and sharpness (Fig. 2b). Based upon the fact that the Young's Modulus of the sample is independent of these manipulations, we reasoned this would allow the model to learn which features were the most informative while preventing overfitting. Consistently, we found that the model generalized much better to validation data when transformations were applied to the training data.

In the final model, we utilized both the DAPI and collagen channels. Given that dead cells are typically quite soft when probed using AFM, the additional information from the DAPI stain helped the model to learn and was included in our imaging studies. We also natural log-transformed elasticity measurements prior to training to alleviate the influence of outliers. At training completion, the correlation of predicted to actual Young's Moduli values was 0.689 (pearson R value; averaged across 25 trained models) when trained over 100 epochs (Fig. 2c, d), which performed significantly better than predicting elasticity based on the intensity of collagen and DAPI alone (multivariable (MV) regression line using all training and validation samples; r = 0.574; p-value (CNN over MV) = 5.23e−10). The validation data is predicted more accurately than the training data due to the transformations applied to the training dataset. Saliency maps indicating image regions that contribute to tissue stiffness demonstrated that the trained models were able to incorporate morphological information from nuclei as well as collagen when predicting stiffness (Fig. 2e)[54–57].

## Generation of STIFMaps

We next applied our trained CNNs to predict the elasticity of normal human and human breast cancer tissue sections across a region of interest using STIFMap. We achieved this objective by segmenting the images into squares matching the input dimensions of the neural network and predicting the Young's Moduli for each square (Fig. 3a, Methods). We then colorized the original images to correspond to the predicted stiffness of each point (Fig. 3b). To validate the performance of these stiffness predictions, tissues were immunostained for two established markers of cellular mechanosignaling, activated β1 integrin and phospho-Myosin Light Chain 2 (pMLC2) that are typically increased in cells in response to a stiff ECM[58,59]. We used the predicted STIFMaps to evaluate the correlation between expression of these markers and stromal tissue elasticity (Fig. 3c). Since a large proportion of the ECM is not directly in contact with cells, we looked at the 99th percentile of stain intensity for each percentile of ECM elasticity (Fig. 3d, Supplementary Fig. 4a, b). This allowed us to remove low-intensity pixels where there were no cells or stain present. The intensity of both mechanosignaling markers positively correlated with the predicted Young's Modulus of the local tissue region (Fig. 3e), but not with the intensity of collagen or DAPI alone (Fig. 3f). We also applied a mask to better identify pixels located at the cell-ECM interface and observed the same trend (Supplementary Fig. 4c, d). The findings indicate that STIFMap can accurately identify mechanical 'hotspots' within human breast tumor tissue sections, thus providing an additional layer of information about the mechanical landscape of human breast tissue that was not previously possible.

## Utilizing STIFMap with formalin-fixed paraffin-embedded tissue

FFPE tissues are frequently used for clinical analysis because this approach preserves cell and tissue morphology. Unlike cryopreserved tissues, which are needed for traditional AFM analysis, FFPE tissues are more readily available for research analysis and clinical translational studies. However, FFPE tissues are highly cross-linked due to formalin-fixation, and thus unsuitable for accurate stiffness measurements by AFM. Accordingly, we asked if STIFMap could predict the elasticity of the original, unfixed tissue samples based solely on collagen

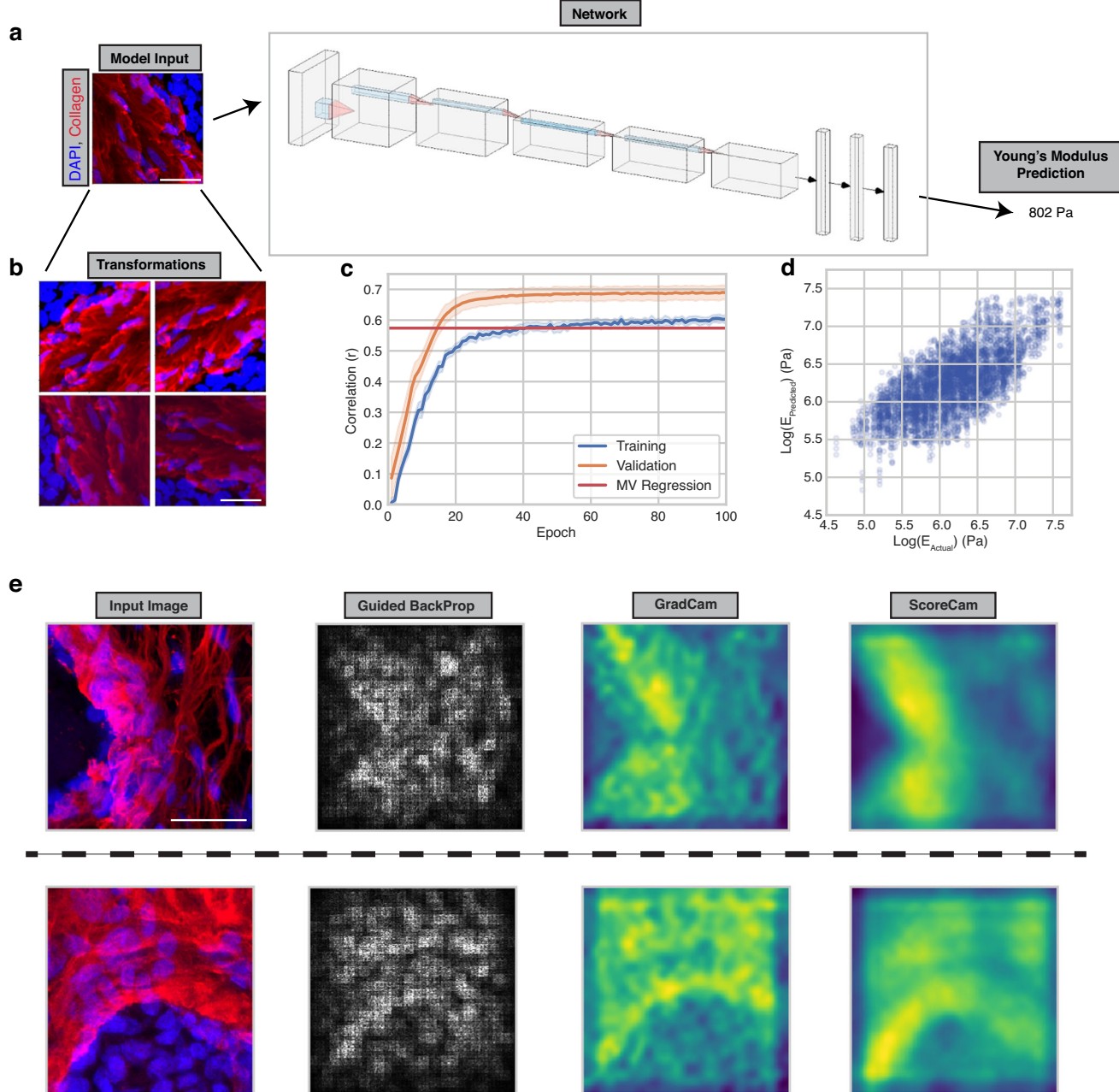

**Fig. 2 | A convolutional neural network predicts the Young's Modulus of tissue.**
**a** Example input-output relationships to the network with diagram depicting connectivity of different network layers. **b** Example image transformations to increase the size of the network training dataset. **c** Correlation between model predictions and actual Young's Modulus values for the training (blue line) and validation (orange line) datasets over the course of training. Error bars indicate 95% confidence intervals across 25 trained models. **d** Dot plot of actual versus predicted Young's Modulus values for the validation datasets across 25 trained models. $n = 4768$, Pearson r = 0.687. **e** Saliency maps reflecting image regions that influenced model predictions. 50 Saliency maps were generated obtaining reproducible results. Scale bar, 20 µm. Source data are provided as a Source Data file.

morphology. We stained terminal duct lobular units (TDLUs) from cryo-preserved and FFPE breast tissues with CNA and DAPI. In consultation with a clinical breast cancer pathologist, we detected no discernable morphological differences between the collagen morphology detected using the CNA collagen stain in a patient-matched FFPE versus cryopreserved tissue (Supplementary Fig. 5a). Moreover, we examined whether formalin-fixation or antigen retrieval (AR) could introduce any changes in collagen morphology that could impact STIFMap predictions. While it is not possible to conduct staining within the same tissue prior and post FFPE tissue processing, to address this issue we imaged collagen with CNA staining in cryopreserved tissues before and after 10% formalin-fixation for 1 h (Supplementary Fig. 5b).

We also stained collagen with PS red in FFPE tissues before and after AR (Supplementary Fig. 5c). In both cases, we did not observe any significant morphological differences in collagen architecture. The results indicate that STIFMap can be applied to predict the elasticity of FFPE tissues in which elasticity measurements are not currently possible.

### A stiff, fibrotic collagenous ECM drives an EMT and tumor metastasis in mice

A stiff ECM can foster the growth, survival, and invasiveness of cultured premalignant and tumorigenic breast cancer cell lines by inducing an EMT[60–63]. A stiff, cross-linked collagenous stroma can also

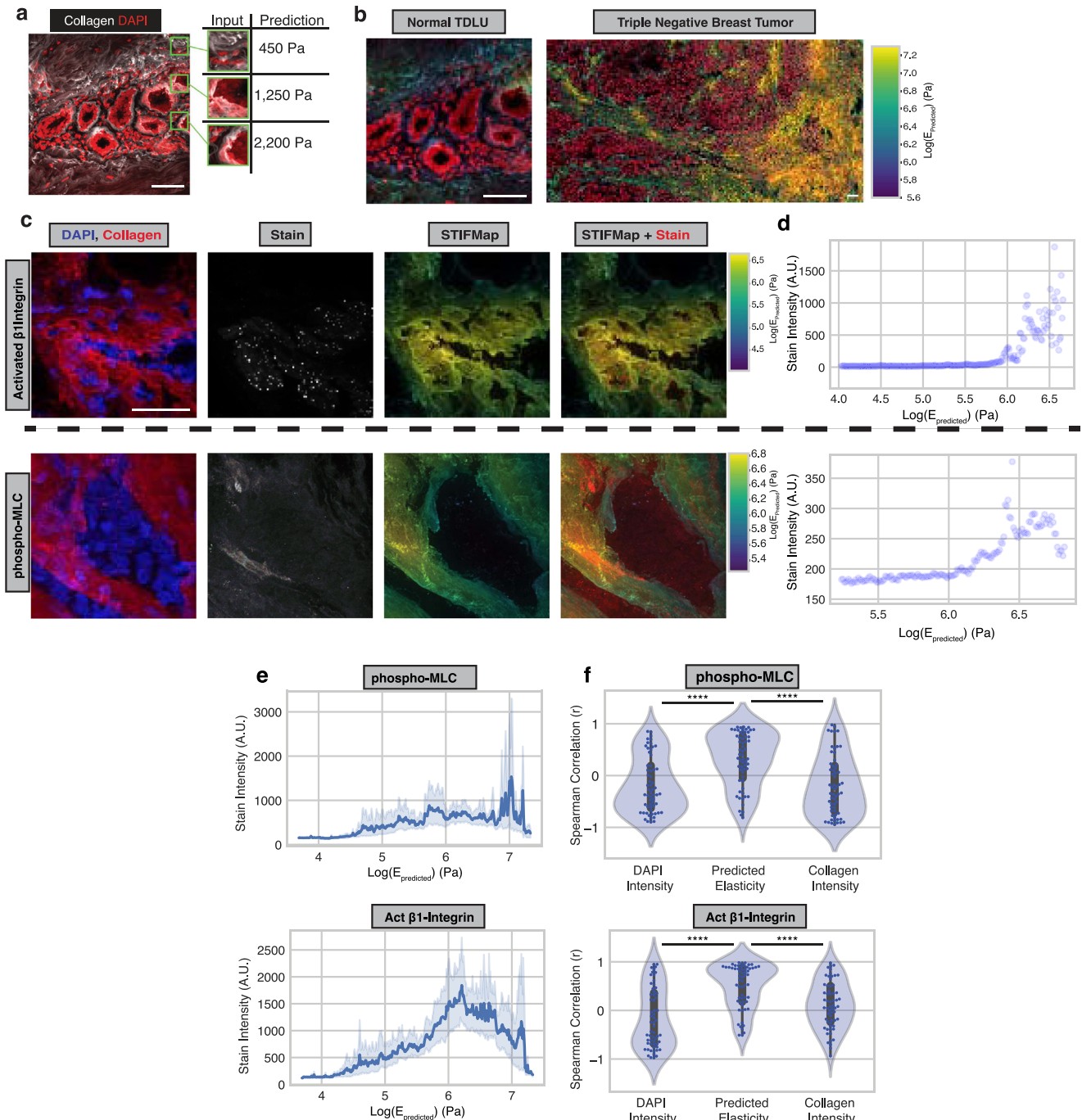

**Fig. 3 | STIFMap predict high elasticity regions within tissues. a** Deconstruction of a CNA- and DAPI-stained image into squares of ~50 × 50 μm. The Young's Modulus of each square is predicted. **b** Elasticity predictions are aggregated and over-laid over collagen to produce the overall STIFMap for both a normal TDLU and triple negative breast cancer. **c** Representative images of immunofluorescent staining for pMLC (top) and activated β1 integrin (bottom) (60 total FOVs from 10 patient samples were imaged). **d** Scatterplots of STIFMap intensity vs stain intensity for each pixel shown in **c** indicating the 99th percentile of stain intensity for each STIFMap percentile. **e** STIFMap percentiles versus the 99th percentile of stain intensity for all acquired fields of view (FOVs). Error bars indicate a 95% confidence interval. $n = 60$ FOVs from 10 different patient tumor samples. Median Spearman r values activated β1 integrin = 0.696, pMLC = 0.364. **f** Violin plots of the Spearman correlation for each FOV comparing the 99th percentile of staining intensity versus percentiles of DAPI, predicted elasticity, or collagen stain intensity. Internal gray bars indicate a Box-plot. The box plots indicate the median, and extreme values. $n = 60$ FOVs from 10 different patient tumor samples. p(Elasticity vs DAPI; p-MLC) = 2.60E−6. p(Elasticity vs Collagen; p-MLC) = 5.28E−6. p(Elasticity vs DAPI; beta1-integrin) = 2.17E−8. p(Elasticity vs Collagen; beta1-integrin) = 2.36E−5. Scale bar, 50 μm. Statistical analyses were performed using two-sided Mann–Whitney U test, ****$P < 10^{-5}$. Source data are provided as a Source Data file.

induce an EMT to promote tumor aggression and metastasis in vivo in experimental transplant murine models of mammary cancer[64]. Nevertheless, there is currently no evidence to directly implicate a stiff, fibrotic stroma in human breast cancer progression and metastasis, nor any published studies that link this phenotype to induction of an EMT. Therefore, to directly test whether a stiff stroma could drive the aggressiveness and metastatic behavior of human breast cancers, and to determine if this is linked to an EMT, we manipulated HER2+ human breast cancer patient-derived xenografts (PDX) in vivo. We reasoned that PDXs are a model that more closely mimics the heterogeneous

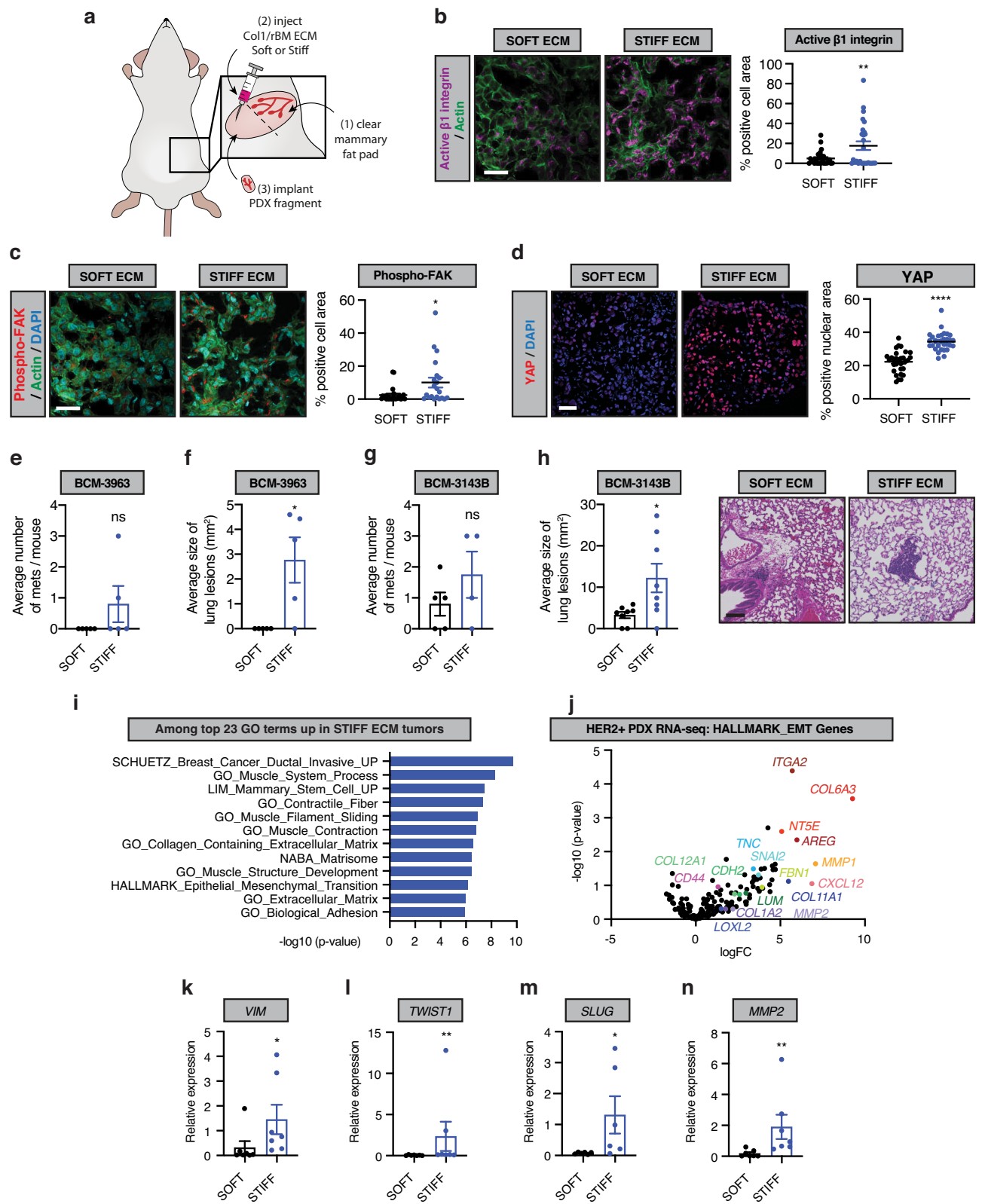

phenotype of human breast tumors[65,66]. To assess this, we implanted three independent HER2+ human breast cancer PDXs (BCM-3963; BCM-3143B, HCI-012) embedded within control (SOFT; 140 Pa) and non-metabolizable L-ribose cross-linked (STIFF; 1200–2000 Pa) collagen gels into the fat pads (orthotopic) of NOD/SCID mice and monitored the impact on tumor phenotype and behavior (Fig. 4a). Immunofluorescence analysis revealed a significant increase in activated β1 integrin, phospho-Y397 focal adhesion kinase activity ($^{Y397}$FAK) and yes-associated protein (YAP) in the PDX tumors that developed within the stiffened collagen gels (Fig. 4b–d). We observed an increase in tumor outgrowth in all three independent HER2 + PDX tumors embedded within the stiffened collagen gels (Supplementary Fig. 6a–c). Markers of growth factor receptor signaling, as indicated by elevated phosphorylated MAP Kinase (pERK; Supplementary Fig. 6d),

**Fig. 4 | Matrix elasticity associates with EMT in PDX models of HER2+ breast cancer. a** Schematic showing the strategy for implantation of HER2-positive patient-derived xenograft (PDX) breast cancer tissues. Representative images of immunofluorescent staining of active β1 integrin (**b**), phospho-FAK (**c**) and YAP (**d**) in SOFT or STIFF HER2-positive PDX tumors (left). Scale bar, 50 μm. Quantification of average active β1 integrin (**b**), phospho-FAK (**c**) and YAP (**d**) positive cell or nuclear area for all HER2-positive PDX tumors (right). SOFT; $n = 6$, STIFF; $n = 6$. **e** Average number of lung metastases for mice bearing BCM-3963 PDX tumors in SOFT and STIFF ECM stroma as determined by histological analysis. SOFT; $n = 10$, STIFF; $n = 10$. **f** Average size of the metastatic lesions corresponding to the analysis in **e**. **g** Analysis as in **e** for mice bearing BCM-3143B PDX tumors. SOFT; $n = 10$, STIFF; $n = 10$. **h** Analysis as in **f** for metastatic lesions corresponding to the analysis in **g** (left). Images of lung metastases for mice bearing BCM-3143B PDX tumors in SOFT and STIFF ECM stroma (right). Scale bar, 100 μm. **i** Gene ontology (GO) terms

from among the top 23 most significantly upregulated, using RNAseq data derived from all HER2-positive PDX tumors generated in SOFT ($n = 9$) and STIFF ($n = 9$) ECM stroma ($n = 3$ for each PDX and condition). **j** Volcano plot of p-value (-log10) vs. log fold change (logFC) for gene expression from the HALLMARK_epithelial-to-mesenchymal transition gene set for RNAseq data of HER2-positive PDX tumors developed in SOFT and STIFF ECM stroma. **k**–**n** qRT-PCR arrays designed to examine EMT related gene expression to analyze RNA isolated from PDX tumors in SOFT and STIFF ECM stroma. SOFT; $n = 7$, STIFF; $n = 7$. Bar plots for the average relative expression of the indicated mesenchymal genes. All graphs are presented as mean +/− S.E.M. Statistical tests used were two-sided Mann−Whitney U test (**c**, **f**, **k**–**n**) and two-sided unpaired t-test (**b**, **d**, **e**, **g**, **h**), *$P < 0.03$, **$P < 0.002$, ***$P < 0.0002$, ****$P < 0.0001$, ns non-significant. Source data are provided as a Source Data file.

---

indicated that tissue tension and integrin signaling promoted tumor cell growth. Histopathological assessment indicated that all HER2 + PDX tumors were of high histological grade however, there was more extensive necrosis in the stiff tumors (STIFF; 58.8% and SOFT; 21.7% necrotic tumor area). There was also a greater number of larger metastatic lesions quantified in the lungs of the mice harboring the stiff tumors (Fig. 4e–h, Supplementary Fig. 6e,f). Consistent with a relationship between a stiff stroma, breast tumor aggression, and induction of an EMT, RNAseq analysis revealed a significant elevation of the 'Hallmark Epithelial Mesenchymal Transition' pathway in STIFF PDX tumors (Fig. 4i, j, MSIGDB pathway M5930). RT-PCR analysis validated the stiffness induction of the expression of several of the EMT genes including Vimentin (*VIM*), *TWIST1*, *SNAI2* (*SLUG*), and *MMP2* (Fig. 4k–n, Supplementary Fig. 6g–l). These findings demonstrate that a stiff stroma induces integrin and growth factor receptor signaling and drives tumor progression and metastasis of human breast tumor PDXs in vivo. The data also implicate stromal stiffness-dependent induction of an EMT in this phenotype.

### STIFMap link stromal elasticity to EMT in patient tumors

Having established that high ECM tension can drive the progression and metastasis of HER2 + PDX breast tumors in association with induction of an EMT, we next applied STIFMap to look for clinical evidence supporting this relationship. We previously showed using AFM and immunofluorescence analysis that the more aggressive triple-negative breast cancer (TNBC) and HER2+ human breast cancer sub-types have more cells with activated β1 integrin and develop a stiffer invasive front[21,22]. We applied STIFMap to explore if there was a sig-nificant association between stromal tension and EMT markers in clinical FFPE samples of TNBC and HER2+ breast tumors. We first looked within patient transcriptomic data and found that expression of collagen genes highly correlated with expression of EMT genes (Fig. 5a, b). Collagen genes were removed from all gene sets to not bias this analysis. We then stained TNBC FFPE tissue sections for Twist1, ZEB1 and SLUG, transcription factors induced by a stiff stroma pre-viously implicated in EMT[61,67–69]. We detected expression of Twist1, ZEB1 and SLUG and found their levels to be positively correlated with the predicted elasticity of the interstitial collagens in the stroma, but not individually with total collagen or DAPI intensity (Fig. 5c–f). A similar trend was also observed across whole-slide images of TNBC tissues (Supplementary Fig. 7a, b). To determine the broader relevance of these clinical findings we next applied STIFMap to a cohort of HER2+ breast tumors with associated clinical follow-up data[70]. We co-stained these FFPE tissue sections with HER2 and ZEB1 as well as with CNA to stain stromal collagens. To facilitate our analysis a pathologist anno-tated tumor regions within each whole-slide image. The predicted tissue elasticity from STIFMap positively associated with ZEB1 stain intensity, but not with HER2, when compared to the correlation with collagen intensity alone (Fig. 5g, h, Supplementary Fig. 7c, d). We then looked at the spatial autocorrelation of each stain within each tissue by

calculating Moran's I, which revealed a trend showing that greater clustering of HER2, ZEB1, and elasticity associated with metastatic recurrence (Fig. 5i). This observation is consistent with worse overall survival among patients with high expression of EMT and collagen gene expression signatures (Fig. 5j, k). These findings demonstrate a spatial link between high stromal collagen elasticity and biomarkers of EMT in both TNBCs and HER2+ human breast tumors. Together with our PDX findings, these data link EMT to ECM stiffness and implicate tension-induced EMT in human breast tumor metastasis. The findings also suggest a stiff ECM could promote tumor aggression and com-promise breast cancer patient outcome.

## Discussion

Here we present a method we term Spatially Transformed Inferential Force Map (STIFMap), which permits the spatial resolution and quantification with micron-resolution of the mechanical hetero-geneity of the collagenous stroma within normal and tumor human breast tissues. The method works on both cryopreserved and FFPE tissues and employs a quick, inexpensive staining protocol via CNA and DAPI. The approach permits simultaneous quantification of the heterogeneous tension landscape of the stromal ECM together with standard biomarker immunostaining approaches, and could further be integrated with spatial proteomics and genomics as well as pro-tein marker multiplexing protocols[24–26]. Although methods do exist to broadly quantify tissue elasticity across a tissue section, they do not provide high-resolution spatial information[35]. AFM is a technique that directly probes tissue elasticity at the single cell scale[71]. However, standard AFM methods are not high-throughput, require fresh or cryopreserved tissue, rely upon specialized equipment and opera-tors, are time-consuming, and only collect sparsely spaced data points over focused sections of a tissue[33, 34]. In the absence of the AutoAFM algorithms presented herein, it is also challenging to overlay AFM data with biomarker staining. Moreover, the use of cryopreserved or fresh tissue compromises the ability to simulta-neously conduct spatial genomic, transcriptomic, or proteomic analyses. STIFMap overcomes current shortcomings of conventional AFM methods and can rapidly annotate the elasticity landscape of whole tissue sections with a simple collagen stain. The method is also amenable to FFPE tissues thereby expanding the scope and applica-tion of the method. Indeed, using STIFMap we were able, to the best of our knowledge, to link for the first time in clinical specimens of human breast cancer, tissue mechanics with indicators of mechan-osignaling and biomarkers of an EMT previously implicated in tumor aggression[36,37]. The results highlight the potential utility of using STIFMap to quantify stromal biophysical features to predict tumor behavior and ultimately patient outcome.

We and others showed both in vitro and in experimental models in vivo that a stiff ECM increases integrin mechanosignaling to foster tumor cell growth, survival, invasion and metastasis, and that this is accompanied by induction of an EMT[60,61,72]. Here we demonstrate,

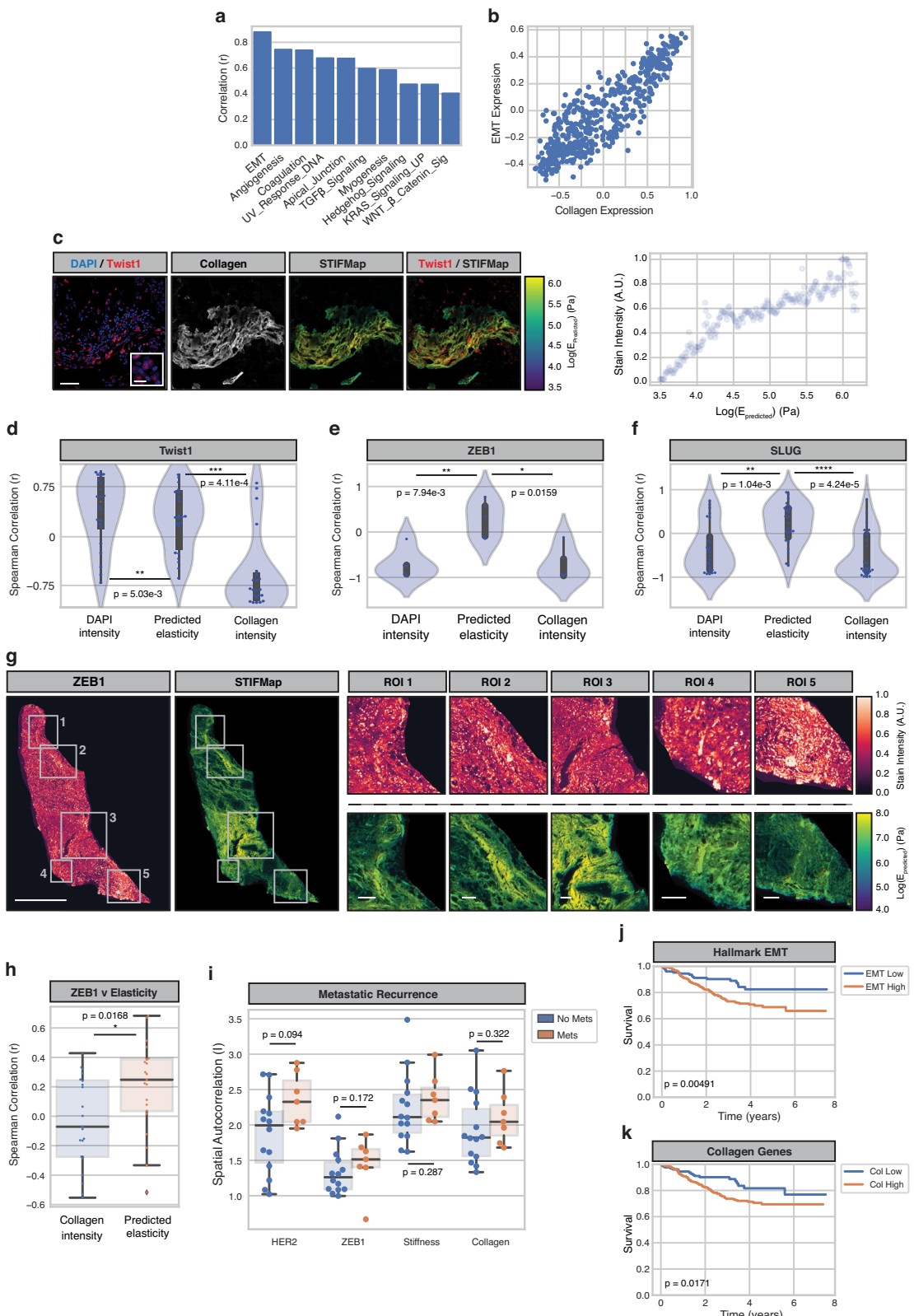

using HER2-positive breast tumor PDX specimens, that a stiff stroma can induce an EMT in vivo and that this is accompanied by metastasis. By correlating stromal tension with biomarkers of an EMT in human clinical specimens of TNBC and HER2+ breast cancer, we found clinical evidence that such a relationship also exists within human tumors, thereby providing validation of the experimental manipulations. Furthermore, we showed that expression of either collagen genes or an EMT signature associates with significantly worse patient outcome.

The findings thereby link stromal tension to human breast cancer progression and directly implicate induction of an EMT in this phenotype. Although STIFMap provides researchers with a versatile tool to explore the role of stromal stiffness in clinical specimens, additional studies will be necessary to further clarify mechanisms through which a heterogeneously stiff ECM drives an EMT in tumors. Moreover, more work will be needed to assess the clinical relevance of stromal stiffness on patient outcome.

**Fig. 5 | EMT markers spatially overlap with high tension matrix and associate with poor survival in patient tumors. a** Pearson correlation between GSVA scores for collagen genes and hallmark pathway genes in the Nuvera dataset. **b** Scatterplot of GSVA scores for collagen genes and hallmark EMT genes. Each point represents one patient. $N = 508$ patients. Pearson r = 0.880. **c** FOVs for Twist1 staining within FFPE tumors (left). Scale bar, 50 μm. Inset scale bar, 10 μm. Scatterplots of STIFMap intensity vs stain intensity for each pixel shown in (i) indicating the 99th percentile of stain intensity for each STIFMap percentile (right). $N = 22$ imaged FOVs. **d**–**f** Violin plots of the Spearman correlation for each FOV comparing the 99th percentile of staining intensity versus percentiles of DAPI, predicted elasticity, or collagen stain intensity. Internal gray bars indicate a box-plot. $N = 22$ Twist1 FOVs, $n = 5$ ZEB1 FOVs, and $n = 25$ SLUG FOVs. **g** Representative whole slide image (WSI) and regions of interest (ROIs) of ZEB1 stain with STIFMap in HER2+ breast cancer

cohort. Scale bar (WSI), 1 mm. Scale bar (ROIs), 100 μm. $N = 21$ patient tumor samples. **h** Spearman correlation for each whole tissue section comparing the 99th percentile of staining intensity versus percentiles of predicted elasticity and collagen stain intensity. $N = 21$ patient tumor samples. **i** Box and whiskers plots to show the association between metastatic recurrence and spatial autocorrelation (Moran's I) for tissue markers and STIFMaps in the HER2+ breast cancer cohort ($n = 84$ breast tumors). Kaplan–Meier curves comparing survival between the upper and lower quartiles of EMT (**j**) and collagen (**k**) GSVA scores within the Nuvera cohort. $N = 127$ patients in each group. Boxes denote 25th to 75th percentile with median line. Whiskers mark the minima and the maxima excluding outliers. Statistical analyses were performed using two-sided Mann–Whitney U test (**d**–**f**, **h**, **i**) and logrank test (**j**, **k**), *$P < 0.05$, **$P < 0.01$, ***$P < 0.001$ ****$P < 10^{-5}$. Source data are provided as a Source Data file.

There is growing interest in the application of artificial intelligence methods to classify clinical histological images[73–75]. While early deep learning algorithms focused on routine tasks such as nuclei segmentation, current state-of-the-art algorithms are beginning to rival pathologists at tasks such as tumor grading and cancer detection[73]. Moving beyond what pathologists are able to detect, some new algorithms are even able to predict tumor recurrence and invasive potential in cohorts for which there is currently no means available for evaluating risk to progression[76]. Notwithstanding these advances, a number of caveats hinder development in this area such as suboptimal network architectures, the requirement for large image datasets of annotated samples, the immense computational processing power necessary to train highly sophisticated models, overfitting data that is generated by only one individual or group, and the difficulty in interpreting why deep learning models classify results in one group or another[77]. Nevertheless, improvements in deep learning such as neural architecture search to find more optimal networks and advancements in computational power continue to make computational pathology more mainstream and accessible in the clinic. While there are still issues to overcome, deep learning algorithms appear to be the future of histopathological analysis and tissue classification[78].

Tumors are highly heterogeneous at the genomic, transcriptomic, and proteomic levels[1,4]. Regions within human and murine tumors have been identified in which immune infiltration, cancer cell metabolism, and stress response pathways exhibit diverse phenotypes. Given that patient prognosis and outcomes have been linked to genomic heterogeneity, as well as variability in immune infiltration and hormone receptor expression, it is perhaps not surprising that there is a growing interest in understanding the relevance of and drivers of ITH[79]. In this regard, the level of tissue fibrosis also predicts patient outcome and recent data suggest the level and organization of tissue collagens and stromal stiffness varies widely within a patient's tumor[21,80,81]. Yet, to date there are no tools with which to spatially resolve the mechanical stromal heterogeneity within a tumor and none that are amenable to scanning across a full tissue section of a tumor. With STIFMap, it is now possible to evaluate the association between a biomarker or pathway of interest and the local and heterogeneous elasticity of the collagen-rich stroma within a given normal or malignant tissue. Moreover, the STIFMap method can be combined with spatial sequencing, in situ gene expression, metabolomics, and even proteomics to allow for unbiased screening of correlations between molecular heterogeneity and mechanically regulated pathways in clinical samples. Accordingly, STIFMap opens the door for clinicians and translational researchers to explore the impact that tissue elasticity has on cellular phenotypic variability in healthy and diseased tissues.

## Methods
### Ethical statement
Our research complies with all relevant ethical regulations. The animal experiments were performed in accordance with the guidelines from the Institutional Animal Care Use Committee (IACUC) protocols,

#AN133001 and #AN179766, which adhere to the NIH Guide for the Care and Use of Laboratory Animals. Within our approved animal study protocol, no tumor exceeded the maximal tumor size of 1.5 cm. Human breast tissue specimens were collected prospectively from consenting patients. Human tissues were stored and analyzed in accordance with the procedures outlined in the Institutional Review Board (IRB) Protocol #10-03832, approved by the UCSF committee of Human Resources and the Duke University IRB Pro00054515.

### Atomic Force Microscopy (AFM)
AFM measurements were performed using an MFP-3D BIO Inverted optical AFM (Asylum Research, Santa Barbara, CA) mounted on a Nikon TE2000-U inverted fluorescent microscope (Melville, NY) and placed on a vibration-isolation table (Herzan TS-150). Silicon nitride cantilevers were used with a nominal spring constant of 0.06 N m$^{-1}$ and a borosilicate glass spherical tip with 5 μm diameter (Novascan Tech). Cantilevers were calibrated using the thermal fluctuation method and verified by probing polyacrylamide gels of known elasticity. The specimens used were 20 μm thick OCT-embedded frozen human breast tissue sections thawed and equilibrated to room temperature by immersion in PBS for 5 min. Thawed sections were immersed in PBS containing phosphatase inhibitor (GenDEPOT Xpert #P3200-001), protease inhibitor cocktail (GenDEPOT Xpert # P3100-001), 5 μg/mL propidium iodide (ACROS, Cat# 440300250), and 3 μg/mL of CNA35-OG488. Specimens were indented at 2 μm per second loading rate. The Young's Moduli of the samples were determined by fitting force curves with the Hertz model using a Poisson ratio of 0.5.

### AFM force plot fitting algorithm
AFM force plots were post-processed to obtain Young's Moduli using a homemade algorithm (see GitHub repository). Briefly, force plots were smoothed using a moving average convolution across 100 datapoints to remove noise and then baseline-corrected using the first third of the AFM indentation curve. The contact point of each force curve was estimated as the point at which the derivative of the force curve increased above an empirically-determined threshold. Then, a more precise contact point was determined by applying a minimization function to fit a flat baseline plus a 1.5-power curve (Hertz Model) onto the AFM data using the estimated contact point as an initial guess. With the contact point determined, the Young's Modulus was calculated to minimize the squared error between the AFM data and the fit curve.

### AutoAFM design
The AutoAFM assembly's function is to ensure proper alignment of the motor relative to the microscope stage's adjustment knob in order to allow the motor to accurately control the knob's rotation and thus the movement of the microscope stage. It does this by supporting the weight of the motor and controlling its position, while allowing the motor to freely slide along its shaft axis. It also allows the operator to fine-tune the motor's position and orientation in space, ensuring good

alignment with (and therefore accurate control of) the stage's adjustment knob.

The assembly has three main components: the Stage Frame, the Motor Frame, and the Motor Bracket. The motors screw into the Motor Brackets via the four Motor Screws. The Motor Bracket sits in the Motor Frame and is pulled down against the Bracket Adjusters by the springs hooked around the Tensioning Pins. By turning the Bracket Adjusting Screws, the Bracket Adjusters can be individually moved forward and backwards, adjusting both the pitch and the roll of the Motor Bracket relative to the axis of the motor shaft. This allows easy manual adjustment of the motor to ensure good alignment between the motor shaft and the stage knob. The Stage Frame is hooked over the lip of the microscope stage, enclosing the stage's fine adjustment knob (not shown), and is able to slide freely along the edge of the stage. The Alignment Rods are press-fit into the Stage Frame and slip-fit into the Motor Frame, allowing the Motor Frame to slide freely towards and away from the Stage Frame along the motor shaft axis. The Motor Coupling joins the motor shaft to the Knob Adapter, which is screwed into the fine adjustment knob via the Adapter Screw. The Y Stage Frame has rollers attached to reduce friction with the AFM stage as it slides side-to-side.

Motor mount components were 3D printed on either a Prusa MINI + or LulzBot Mini 2 with PETG. A high infill was used for ease of sanding. Some dimensions were slightly oversized so that they could be gradually sanded to fit snugly. See https://github.com/cstashko/AutoAFM/STL for a full list of part STL files. Other components were ordered from McMaster-Carr. See supplementary table 1 for a full Bill of Materials. Motors were driven via an Arduino Mega 2560 Rev3 Classic microcontroller interfacing with RAMPS 1.4 (https://reprap.org/wiki/RAMPS_1.4).

AutoAFM operates by moving the AFM cantilever to user-defined positions and acquiring an AFM force curve at each point. Within MicroManager, the user draws a path via the Freehand Line or Segmented Line tools and specifies the step size between points as well as the initial cantilever position[82]. For each point, the motors attempt to move to the desired location. An image is taken at the new location and stitched together with existing images using Phase Cross-Correlation to determine the actual AFM movement that occurred[83]. If the AFM cantilever is within tolerance of the desired position, then a force curve is acquired. Otherwise, the motors make additional movements until the cantilever position is within tolerance (Fig. 1c). At completion, AutoAFM returns force curves and positions for each of the user-specified points. Full code and a complete pipeline for AutoAFM acquisition is available via https://github.com/cstashko/AutoAFM/.

### Polyacrylamide hydrogels
Polyacrylamide (PA) hydrogels of varying rigidities were prepared as described[84,85]. Briefly, PA gels of specified rigidities were mixed according to previously reported ratios[85], omitting 1% potassium persulfate (PPS). Solutions were degassed for 20 min, then PPS was added and 300 μL was quickly deposited onto a Rain-X™-coated 60 mm coverslip and sandwiched with a glutaraldehyde-activated coverslip. After one hour of polymerization, Rain-X™-coated coverslips were removed. Gels were stored in PBS. For shear rheology studies, gels were cast directly onto the baseplate of an AR 2000 rheometer (TA Instruments) and immediately compressed to a barrel shape using a 25 mm diameter probe. Gels polymerized for two hours at room temperature with a 1% applied strain at a frequency of 1 rad*s⁻¹. The shear modulus was measured for 2 h while following a liner relationship. A Poisson's Ratio of 0.457 was used when calculating the Young's Modulus of PA gels[86].

### Micropatterned substrates for AFM control studies
Photolithography and soft lithography were used to generate polydimethylsiloxane (PDMS, Dow Silicones Corporation) substrates with defined ridge topographies (15 μm height, 100 μm length, and widths ranging from 12 μm to 0.5 μm) for use in AFM control studies. Briefly, a silicon wafer was plasma treated (5 min, Harrick Plasma) and a 2 μm tall adhesive layer of SU-8 2002 (Kayaku Advanced Materials) was cast onto the wafer surface using a spin coater (Laurell Technologies). A 4 cm square was UV patterned onto the adhesive layer using the PRIMO photopatterning system (Alvéole). A second 15 μm layer of SU-8 2010 (Kayaku Advanced Materials) was then cast onto the wafer, and ridge arrays (100 μm length, 12 μm–0.5 μm width, 30 μm spacing) were subsequently photopatterned. Patterned wafers were developed using propylene glycol monomethyl ether acetate (PGMEA, Sigma-Aldrich), cleaned with isopropyl alcohol (IPA, Sigma-Aldrich), and dried with n-pentane (Acros Organics) and N2 gas. PDMS was poured over wafer patterns and cured for 15 min at 100 °C to generate a negative mold. The negative mold was silanized overnight by vapor deposit of trichloro(1H,1H,2H,2H-perfluorooctyl)silane (TFPS, Sigma-Aldrich). A second layer of PDMS was poured over the silanized negative mold, and a glass coverslip was applied to sandwich the layer of PDMS. This PDMS was cured at 100 °C for 20 h to generate a positive mold of ridges adhered to a glass coverslip, which was then used for AFM control studies. Fluorescent beads 1.0 μm in diameter (Thermo-Fisher F8814, 1:500 dilution in water) were allowed to settle into the PDMS for visualization purposes during AutoAFM.

### Image registration
Registration for images with the same scale and orientation was computed using the phase_cross_correlation function from skimage[83]. For images with different scales and orientations, transformations were found by applying a Fourier Mellin Transform[42]. Briefly, images were applied with a band-pass filter followed by a Hanning Window. Images were then transformed using a Fast Fourier Transform (FFT) and magnitudes were log-polar transformed. Translations between these transformed images were calculated using phase_cross_correlation, which can then be used to calculate the rotation and scaling differences in the original images.

### Neural network design
Networks were designed in Pytorch. Input images were used of size $224 \times 224 \times 3$ pixels in which the three channels are DAPI, CNA, and a layer of zeros, which was incorporated for ease of use with existing Python machine learning image loading functions. Within the training dataset, images were loaded with size $448 \times 448 \times 3$ pixels, transformed via random rotation of 0–180 degrees, randomly flipped with probability 0.5, received adjustments to brightness, contrast, and sharpness, and cropped to $224 \times 224 \times 3$ to remove any zero pixels resulting from rotation. A mini batch size of 16 was used throughout. For each round of training, samples were randomly split by patient with a training:validation ratio of 0.8:0.2 so that the validation dataset only contained samples from patients that were not included in the training set. Mean squared error was used as a loss function. Learning rate and weight decay were set at 4e−6 and 4e−7, respectively. A dropout rate of 0.5 was used for fully-connected layers. Networks were trained for 100 epochs since this is when accuracy for the validation set converges. Code for network visualizations was modified from https://github.com/utkuozbulak/pytorch-cnn-visualizations#gradient-visualization.

### Human breast tissues
All human breast tissue specimens were collected prospectively from consenting female patients (all patients provided written informed consent prior to surgery) undergoing surgery at the University of California, San Francisco, (UCSF) or Duke University Medical Center between 2010 and 2020. All patients were treatment naïve when clinical samples were collected. Only samples of invasive ductal carcinoma that were centrally verified as HER2+ or TNBC were analyzed.

For the clinical cohort of HER2+ samples, untreated core needle biopsies were compared, and metastatic recurrence data was collected. Samples were stored and analyzed with deidentified labels to protect patient data in accordance with the procedures outlined in the Institutional Review Board Protocol #10-03832, approved by the UCSF Committee of Human Resources and the Duke University IRB (Pro00054515). Tissue specimens were flash frozen in OCT (Tissue-Tek) by slow immersion in liquid nitrogen or placement on dry ice and stored at −80 °C until ready for sectioning. H&Es were performed on an adjacent slide and were scanned using a ZEISS Axio Scan.Z1 digital slide scanner equipped with CMOS and color cameras, 10×, 20× and 40× objectives. H&E-stained tissues were assessed by a pathologist (A.J.I.) to identify regions of interest for AFM measurements.

## CNA35-OG488 transformation and purification

pET28a-EGFP-CNA35 was a gift from Jan Liphardt[52] (Addgene plasmid # 61603). CNA35 was expressed and purified as previously described[87]. Briefly, bacteria were incubated with 5 mL of 2YT media + 100 µg/mL ampicillin + 25 µg/mL kanamycin + 1% wt/v glucose overnight at 30 °C in a shaking incubator. The next day, the culture was diluted in 50 mL of 2YT media + kanamycin for three hours. The culture was then centrifuged, and the supernatant discarded. Then, the sample was digested for cell wall removal for 30 min using 500 µL of lysis buffer (50 mM Sodium Phosphate dibasic, 20 mM Imidazole, 300 mM NaCl, pH 8.0) supplemented with 0.125 mg Lysozyme and 1 mM DTT. The sample was sonicated and centrifuged, then CNA35 was isolated from the solubilized supernatant via affinity chromatography (Qiagen Ni-NTA Agarose) according to manufacturer's instructions. The purified protein was supplemented with 40% glycerol and stored at −20 °C. Under typical isolation conditions we obtained a final concentration of approximately 1.5 mg/mL CNA35.

## Collagen/rBM hyrdogels with orthotopic implantation of tumor cells

Rat tail collagen-1 (High concentration, Corning, Cat. #: 354249) was incubated with 0.1% acetic acid (non-crosslinked; SOFT) or 0.1% acetic acid with 500 mM L-ribose (Chem Impex International, Cat. #: 28127) (cross-linked; STIFF) for at least 10 days before preparation of Col1/rBM hydrogels for orthotopic implantation of tumor cells or tumor fragments[64,88]. Col1 mixtures were then combined with basement membrane extract (R&D Systems, Cultrex BME, type 2, Pathclear, Cat. #: 3532-005-02) (20% final volume), PBS, and 1N NaOH to a slightly acidic pH (pH ~6.5) as determined by pH strips. Col1/rBM with and without L-ribose was injected orthotopically into a cleared inguinal fat pad and allowed to set for 3–5 min prior to implantation of a PDX tissue fragment ~2 × 2 mm in size.

## Breast cancer Patient-Derived Xenografts (PDXs)

PDX tissues were obtained from Dr. Alana Welm at the Huntsman Cancer Institute, University of Utah, Utah (HCI-012) or Dr. Michael Lewis at Baylor College of Medicine, San Antonio, Texas (BCM-3143B and BCM-3963)[65,66]. For the PDX study, 2 × 2 cm breast tumor specimens were collected as fresh tissue with immersion in media (phenol red free-DMEM/F12) with 10% charcoal-stripped fetal bovine serum (FBS Benchmark, Cat. #: 100−106) and GlutaMAX (Gibco, Cat. #: 35050-061) supplementation for transportation to the Weaver laboratory at UCSF. PDX fragments were established from frozen and implanted in 3-week old female NOD-SCID (strain: 001303, Jackson Laboratory) immunodeficient mice for maintenance. Once established tumors reached a maximal tumor size of 1.5 cm diameter, mice were sacrificed, and tumor tissue was divided into pieces for formalin-fixation and paraffin embedding, embedding and freezing in OCT, and flash freezing in liquid nitrogen and cryopreservation in 95% FBS: 5% DMSO. Flash frozen tumor pieces were used for RNA and protein isolation for the downstream applications indicated.

## Animals and animal care

Animal husbandry and all procedures on mice were carried out in Laboratory Animal Resource Center (LARC) facilities at UCSF Parnassus in accordance with the guidelines stipulated by the Institutional Animal Care Use Committee (IACUC) protocols, #AN133001 and #AN179766, which adhere to the NIH Guide for the Care and Use of Laboratory Animals. Female NOD/SCID mice were purchased at 3-weeks of age from Jackson Laboratories for orthotopic implantation assays. Mice were sacrificed twelve weeks after injection or at maximal tumor size of 1.5 cm. No tumor exceeded the maximal tumor size of 1.5 cm. Tumors were excised and examined for tumor volume using calipers, histology by H&E of fixed tissue sections, proliferation and growth factor and integrin signaling via immunofluorescence in tissue sections, and gene expression using RNAseq and RT-PCR.

## Monitoring of tumor growth and metastasis

Tumor growth was monitored by palpation and caliper measurement weekly or biweekly. Lung metastases were quantified by counting of surface lesions at time of animal sacrifice, and by examination of histological lung sections stained by H&E. Lungs were scanned using a ZEISS Axio Scan.Z1 digital slide scanner equipped with CMOS and color cameras, 10×, 20× and 40× objectives, and lesion area was determined by tracing metastatic lesions in QuPath[89].

## Quantitative reverse transcriptase-polymerase chain reaction (qRT-PCR)

RNA was prepared from flash-frozen and pulverized mammary tumor tissues using TRIZol reagent (Invitrogen). Reverse transcription reactions were performed using M-MLV reverse transcriptase (Biochain, Cat. #: Z5040002) with random hexamer primers. cDNA was mixed with PerfeCTa SYBR Green FastMix (Quantibio, Cat. #: 95072-05K) for qPCR analysis using an Eppendorf realplex2 epgradient S mastercycler. Thermal cycling conditions were 10 min at 95 °C, followed by 40 cycles of 15 s at 95 °C and 45 s at 65 °C. Melting curve analysis was used to verify primer pair specificity. Relative mRNA expression was determined by the ΔΔCT method with normalization to *GAPDH* or *18S*.

## Quantitative polymerase chain reaction (qPCR) arrays

Human EMT qPCR arrays were purchased from Qiagen (Cat. #: PAHS-021Z), performed as described using RNA from PDX mammary tumors grown in SOFT and STIFF Col1/rBM hydrogels, and analyzed using available product resources from Qiagen. Selected genes were plotted for presentation in Fig. 4 and Supplementary Fig. 6.

## Immunofluorescence

Immunofluorescence was performed using the following specific antibodies: phospho-FAK (Y397) (Cell Signaling Technology, Cat. #: 8556, 1:200), phospho-p44/42 MAPK (ERK1/2) (T202/Y204) (Cell Signaling Technology, Cat. #: 9101, 1:200), Integrin β1, activated (Sigma-Aldrich, clone HUTS-4, Cat. #: MAB2079Z, 1:400), phospho-Myosin Light Chain 2 (Ser19) (Cell Signaling Technology, Cat. #: 3671, 1:200), yes-activated protein (YAP) (Santa Cruz Biotechnology, Cat. #: sc-15407, 1:200), SLUG (C19G7) (Cell Signaling Technology, Cat. #: 9585, 1:200), ZEB1 (E2G6Y) (Cell Signaling Technology, Cat. #: 70512), Twist1 (Abcam, Cat. #: ab50887, 1:500) and ErbB2/HER2 (Abcam, clone 3B5, Cat. #:ab16901). For cryopreserved samples, frozen sections were fixed in 2−4% paraformaldehyde, prior to permeabilization with 1−3% triton-x-100 and incubation with primary antibodies overnight at 4 °C with 3 µg/mL CNA where specified. Sections were then incubated with species-specific secondary antibodies conjugated to different fluorophores (AF-555, −647, Invitrogen). All washes were carried out using Phosphate-buffered saline (PBS) with 0.5% Tween-20 and nuclei and/or actin filaments were counterstained using 4′,6-diamidino-2-phenylindole (DAPI, Cat. #: D1306) and Phalloidin-AF488 conjugate (Thermo Fisher Scientific, Cat. #: A12379), respectively. For FFPE samples,

antigen retrieval was accomplished by boiling sections in 10 mM citrate buffer in a pressure cooker on high power for 3 min. Following blocking with 10% goat serum and 1% BSA in Tris-Buffered Saline (TBS), sections were incubated with primary antibodies overnight at 4 °C with 3 μg/mL CNA. Sections were incubated for 1 h with species-specific secondary antibodies conjugated to different fluorophores (AF-555, −647, Invitrogen). All washes were carried out using TBS with 0.025% Triton X-100 and nuclei were counterstained using DAPI. Images of stained sections were acquired on either a Leica TCS SP5 Confocal microscope, Nikon SoRa Spinning Disk microscope or an inverted Eclipse Ti-E Nikon microscope with CSU-X1 spinning disk confocal (Yokogawa Electric Corporation), 405 nm, 488 nm, 561, 635 nm lasers; a Plan Apo VC 60X/1.40 Oil or an Apo LWD 40X/1.15 Water-immersion λS objective; electronic shutters; a charge-coupled device (CCD) camera (Clara; Andor) and controlled by Metamorph.

## Image analysis

For STIFMap generation, immunostaining images are first resized to the same resolution as the panels used to train the neural networks. Then, the image is decomposed into squares the same dimensions as the network training panels and separated by a user-defined step size that is smaller than the panel side length. The elasticity of each square is predicted using five independently trained models with different brightness, sharpness, and contrast transformations. Since elasticity predictions only apply to panel centers where the AFM cantilever would make contact, the elasticity of pixels between panel centers is inferred using cubic spine interpolation. STIFMaps are depicted as collagen pseudocolored to reflect the predicted elasticity of each position.

Image analysis of percent positive area in PDX samples was performed using ImageJ and QuPath software[89,90]. For comparison, immunofluorescence images were subjected to same-level thresholding based on a determined range of positive fluorescence intensity in each channel and antibody staining panel and the threshold area was expressed as a percentage of whole cell or nuclear area using DAPI staining measured in the same manner.

## RNA-seq library preparation, sequencing, and analysis

RNA was isolated using TRIzol (Invitrogen, Cat. #: 15596018) followed by chloroform extraction. RNAseq library preparation was performed by the Functional Genomics Laboratory (FGL), a QB3-Berkeley Core Research Facility at UC Berkeley. Total RNA samples were checked on a Bioanalyzer (Agilent) for quality and only high-quality RNA samples (RIN > 8) were used. At the FGL, Oligo (dT)25 magnetic beads (Thermofisher) were used to enrich mRNA, and the treated RNAs were rechecked on the Bioanalyzer for their integrity. The library preparation for sequencing was done on Biomek FX (Beckman) with the KAPA hyper prep kit for RNA (now Roche). Truncated universal stub adapters were used for ligation, and indexed primers were used during PCR amplification to complete the adapters and to enrich the libraries for adapter-ligated fragments. Samples were checked for quality on an AATI (now Agilent) Fragment Analyzer. Samples were then transferred to the Vincent J. Coates Genomics Sequencing Laboratory (GSL), another QB3-Berkeley Core Research Facility at UC Berkeley, where Illumina sequencing libraries were prepared. qPCR was used to calculate sequence-able molarity with the KAPA Biosystems Illumina Quant qPCR Kits on a BioRad CFX Connect thermal cycler. Libraries were pooled evenly by molarity and sequenced on an Illumina Nova-Seq6000 150PE S4. Raw sequencing data were converted into fastq format, sample-specific files using the Illumina bcl2fastq2 software on the sequencing centers local Linux server system. RNAseq fastq files were mapped to the primary assembly of the Gencode v33 human genome using Rsubread (version 2.0.1) and counted using feature-Counts. Lowly expressed genes were filtered out if they did not have at least one count per million (CPM) in at least 4 samples. Data

normalization was performed using calcNormFactors in edgeR (version 3.28.1). Gene ontology was performed using Gage (version 2.36.0) with gene lists from MSigDB version 7.2.

## Nuvera dataset analysis

Nuvera patient microarray data was obtained from GSE25066 using GEOquery (v2.60.0)[91]. Expression intensities were normalized between patients using the 'normBetweenArrays' function in the R package limma (v3.48.3)[92]. Gene set enrichment scores were computed using GSVA (v1.40.1) to estimate the abundance of each 'Hallmark' ('H' collection) gene set from MSIGDBR (v7.4.1) as well as a list of the 12 most highly expressed collagen genes[93]. All collagen genes were removed from Hallmark gene sets to prevent artifactually high correlations due to the same gene being included in both sets. Correlations between GSVA scores were plotted in Python using Seaborn (v0.11.2) and Matplotlib (v3.5.1). Kaplan–Meier curves and statistical testing was conducted in Python using the 'lifelines' package (v.0.27.0). All analysis code is available via GitHub repository https://github.com/cstashko/STIFMaps.

## Statistical analysis

Unless otherwise stated, statistical analyses were performed using GraphPad Prism Version 9.1.2 or SciPy Version 1.7.3. Statistical tests used as well as significance is noted in the corresponding figure legends. Tests of normality were used to determine the appropriate statistical test. All independent variables are described in the text with measurements always from distinct samples (biological replicates) unless otherwise stated. All tests are two-tailed unless otherwise indicated.

## Illustrations

The AutoAFM feedback system schematic (Fig. 1c) was created with BioRender.com (licensed to V.M.W.). The AlexNet visualization (Fig. 2a) was created using NN-SVG (http://alexlenail.me/NN-SVG/AlexNet.html). Figure 1a, b and Supplementary Fig. 1a, b were generated with free license software OnShape.

## Reporting summary

Further information on research design is available in the Nature Portfolio Reporting Summary linked to this article.

# Data availability

The authors declare that all data supporting the findings of this study are available within this publication and its Supplementary information. Source data are provided as a Source Data file. PDX RNAseq data has been deposited in NCBI's Gene Expression Omnibus[94] and are accessible through GEO Series accession number GSE179983. Neural networks, training data, stain imaging, and STIFMaps are available at https://github.com/cstashko/STIFMaps and https://data.mendeley.com/datasets/vw2bb5jy99/2. It is also available in Zenodo under the https://doi.org/10.5281/Zenodo.78892270, https://doi.org/10.5281/zenodo.7882270. AutoAFM part files and assembly instructions are available at https://github.com/cstashko/AutoAFM. Source data are provided with this paper.

# Code availability

All code necessary to implement STIFMap is available via the Github repository https://github.com/cstashko/STIFMaps. It is also available in Zenodo https://doi.org/10.5281/zenodo.7882270. AutoAFM code is available at https://github.com/cstashko/AutoAFM. AlexNet Pytorch implementation is from https://github.com/pytorch/vision/blob/main/torchvision/models/alexnet.py and network visualization code modified from https://github.com/utkuozbulak/pytorch-cnn-visualizations. All other code used in the preparation of this manuscript is publicly available from software and commercial sources.

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

## Acknowledgements

We thank Nataliya Korets for care and handling of animals and for tissue histology. We also thank John Eichorst and Austin Edwards within the Biological Imaging Development Center (BIDC) at UCSF for microscopy support, Dylan Romero at the UCSF Library Makers Lab for 3D printing components for AutoAFM, Joanna Ho for hardware engineering consultations, and Ilona Berestjuk for immunofluorescent protocol contributions. We thank Kevin Tharp for technical assistance and discussions. PDX tissues were obtained from Dr. Alana Welm (Huntsman Cancer Institute, University of Utah) and Dr. Michael Lewis (Baylor College of Medicine). RNAseq was conducted by QB3 Genomics (*QB3 Genomics, UC Berkeley, Berkeley, CA, RRID:SCR_022170)* supported by NIH S10 OD018174 Instrumentation Grant. pET28a-EGFP-CNA35 was a gift from Jan Liphardt. AlexNet Pytorch implementation code was obtained from https://github.com/pytorch/vision/blob/main/torchvision/models/alexnet.py and network visualization code modified from https://github.com/utkuozbulak/pytorch-cnn-visualizations. This work was supported by R35 CA242447-01A1, R01 CA222508, and The Mark Foundation for Cancer Research to V.M.W., the National Foundation for Cancer Research A139054 to K.P. and V.M.W., an R35 CA197623 to K.P. and by a CIHR fellowship to M-A.G. We also acknowledge support by an NIH shared equipment grant S10OD028611-01 for use of the Nikon SoRa Spinning Disk Confocal microscope.

## Author contributions

C.S. and V.M.W. conceived and designed the study. C.S. and V.M.W. directed the studies. C.S. performed all AFM analysis and subsequent imaging. C.S. implemented AutoAFM computer vision and STIFMap neural networks and developed image analysis pipelines. C.S. performed RNAseq and gene set enrichment analyses. M-K.H., J.J.N., and C.S. performed IHC staining and imaging for human tissue samples. J.J.N. and M-K.H. completed all animal studies. J.J.N. performed collagen/rBM hydrogel studies in vivo with PDX tissues. J.J.N. and M-K.H. performed H&E and IHC staining and qRT-PCR analysis of mouse tissues. A.J.I. assessed mouse and human tissue pathology. N.P. designed and assembled AutoAFM mounts. L.M. performed PRIMO PDMS fabrication under supervision from M.K. J.N.L. performed isolation of CNA. R.O. contributed technical expertise, AFM analysis insight. E.S.H. and H.R. collected and provided human breast tissue specimens with patient data. M-A.G. performed staining for HER2 and ZEB1 within the clinical HER2+ breast cancer cohort. K.P. provided resources and supervision pertaining to the HER2+ clinical cohort. V.M.W. and C.S. wrote the manuscript with editorial input from all authors.

## Competing interests

The authors declare no competing interests.
