## [Peer review file · Nature Communications]

REVIEWER COMMENTS

Reviewer #1 (Remarks to the Author):

The article by Stashko et al. investigates the possibility of developing an automatized atomic force microscopy (AFM) technique with the intention to obtain reliable spatial information about tissue stiffness. Briefly, they created a motorised stage for AFM, worked with PI-stained samples (to guide imaging and stitch images) and measuring PA gels of defined stiffness before and after each imaging (for calibration). They further overlaid these images with high magnification DAPI stain, used a fluorescent collagen binding protein to stain collagen structures and developed a convolutional neural network expose tissue elasticity onto collagen morphological features.

After validating their system, the authors investigated stiffness associated metastatic potential using human HER2+ PDX samples and further HER2+ or TNBC primary breast cancer samples. Their results suggest the new technique (STIFmap) can predict metastatic potential. Additionally, STIFmaps allowed spatial association of ECM stiffness to EMT gene expression.

This is a well thought and well invested study to make AFM an easy and mainstream technique to measure tissue stiffness. This new technique allows identification of mechanical hotspots where cells interact with collagen rich ECM, therefore will add to spatial gene expression knowledge base in a contextual manner. The article is well written and presented. Figures were informative.

Comments:

- Theoretically, HER2+ tumours should have active Ras pathway and therefore have abundant P-ERK. Is there another and more specific way to assess the effects of ECM stiffness and integrin signalling in PDX modelling? For example, survivin and bcl-2 upregulation or p27 downregulation were shown to be positively regulated by integrin signalling. In addition, YAP/TAZ signalling can be investigated in this context.
- The use of STIFMaps in FFPE is investigated but data provided is limited. I understand morphological features of cryopreserved or FFPE tissues were reflected through CNA collagen staining but to address the problem correctly, tissue stiffness and collagen structure should be assessed before and after FFPE staining and assessing a correlation in a true experimental model.
- Are there any morphological or histopathological differences in mets of PDX tumours embedded in soft or stiff material?
- The authors stained TNBC tissues with ZEB1 and SLUG, quoting an article in melanoma. Breast cancer and melanoma are very different in nature. Nuclear localisation of Twist due upon exposure to stiffer ECM has been reported in breast cancer and accepted as a reason of stiffness associated EMT (Wei et al 2015, Fattet et al. 2020). This aspect of Twist1 (nuclear localisation) cannot be assessed by RNA abundance. It is a good idea to investigate Twist1 expression (rather than SLUG), especially at tumour-stroma interface, along with STIFmap technique in patient materials.

- Can STIFmap technique determine collagen bundling? There are plenty of reports suggesting collagen bundling can affect or determine metastatic potential.

Please italicize Latin phrases (*in situ*, etc)

Reviewer #2 (Remarks to the Author):

In this study, the authors estimate tissue stiffness with a microscopic technique. To the reader it is unclear if the main advance is the biological understanding or the method per se. The whole article is a bit confusing at the first read. Only after reading the whole thing the reader realizes that this is actually mostly an engineering article which reports a new AFM protocol with some data analysis. To me it is unclear who the readership of this article would be, apart from people in the AFM field.

The phrasing of the title is odd. The message should focus on the biological finding, and could mention the tool used to make this finding. Please rephrase.

In the abstract, abbreviations such as AFM are not defined. The abstract is confusing to read. It is again unclear what was the state of the art before the study and what's the novelty and the key finding. The last sentence of the abstract does not seem to be substantiated by data.

The introduction is a bit excessive, trying to link many biological concepts, but in my opinion it should focus more on the topic at hand, the mechanical characterization of tissue.

The results report the outcome of a lot of experimental work, which is not directly related to the topics discussed in the introduction. The results have too much technical details which should be moved to the methods section.

The figures have a high visual quality

Reviewer #3 (Remarks to the Author):

This manuscript proposes an automated AFM system capable of measuring the mechanical distribution of breast cancer tissue slides which combined with other traditional biological and clinical detection and analysis methods, to study the relationship between tissue stiffness heterogeneity and breast cancer tumor aggression behavior. The AFM mechanical measurement data, corresponding CNA and DAPI images are used to train the CNN network to generate the elasticity distribution of tissue sections. The proposed STIFMaps method also has the potential to predict the mechanical distribution of formalin-Fixed paraffin-embedded tissue sections that have lost their intrinsic mechanical information. Experiments show that the STIFMaps obtained by the CNN network is consistent with the information extracted by biological methods and can reveal the relationship between the stromal stiffness of human breast tumor specimens and human breast cancer aggression. This is a very interesting paper and provides a promising way to expand AFM mechanical measurement from cell-scale to tissue-scale. In addition, the STIFMaps solves the problem that traditional AFM cannot perform mechanical mapping of clinical tissue specimens and makes a great contribution to the field of AFM. At the same time, it also provides a versatile tool for clinical to analysis, treat and predict breast cancer. Here are some comments.

1. The accuracy of AFM measurements of mechanical properties of tissue section samples should be considered. The article discussed the case where the cantilever was contaminated. But the tip which is the 5um radius sphere is more easily contaminated and contributes more to the potential bias in the modulus measurement result. The contact area and the tip modulus have a major influence on the modulus calculated from the contact model such as Hertz model. If the surface of the sphere is contaminated, the shape and surface area will change. In addition, if the modulus gap between the sphere and the contamination is large, the bias of measurement result should be considered and discussed. It is recommended that the author characterize the microspheres after multiple measurements or perform secondary calibration on the standard sample to eliminate the influence of tip contamination on measurement results.

2. The surface of the thawed tissue slides sample will not be as smooth as that of the single cell sample. If the fluctuation of the sample surface topography is larger than 5 microns which is the diameter of the sphere used in this article, it is likely that the cantilever beam will touch the sample surface first rather than the sphere. In this case, the obtained force curve cannot be applied to the contact model based on sphere to calculate the sample modulus. Therefore, it is necessary to characterize the surface of the sliced tissue to evaluate the validity of the force curves acquired by the sphere of 5um diameters. Maybe the diameter of the sphere should be increased or the flatness of the tissue sample surface should be further improved.

3. The thickness of the specimens used were 20 μm . The indented depth was 2 μm for each force curve. The substrate effect will affect the accuracy of the Modulus result if the indentation depth is too large. The suitable value of indentation depth is considered as less than 10% of the sample thickness. In this paper, the indentation-thickness ratio is 10%, so I suggested that the substrate effect should be considered and studied by decreasing the indentation depth. Besides, the typical force curves in various positions of the specimen should be provided in supplementary information or extended data. The difference in the obtained mechanical properties results can be evaluated from the force curves.

4. Using STIFMaps to predict the tissue mechanical distribution of Formalin-Fixed Paraffin-Embedded Tissue is one of the contributions in this paper, which will also provide potential contribution to the pathology and clinical medicine. But the authors didn't provide further experimental results to prove the feasibility of this method. It is suggested that the comparison of the modulus between the measured fresh sample and the predicted FFPE sample should be added. This will significantly improve the quality of the article.

5. For the CNN used in this paper, it is mentioned that Alexnet has a better effect than other networks. However, the performance of neural network ResNet and DenseNet in images classification is better than that of AlexNet proposed in 2012. So my question is why AlexNet has better effect than other networks on your task? Regarding the use of AlexNet in the pytorch library, whether a pre-trained model is used (the parameters trained by AlexNet on a large data set are loaded during initialization), and how many images are used during training, and whether there is any overfitting during the training process? (The performance on the train set is much better than the test set).

Rebuttal Comments:

We thank the reviewers for their constructive comments. We have addressed each of their critiques by conducting additional experiments, validations and analysis. We comprehensively revised the text of the manuscript and included additional panels in the figures and updated main and supplemental figures. We include below a detailed point by point response to each of the reviewer's comments. We are confident that the editors and reviewers will be satisfied with our revised manuscript and accept our manuscript for publication in Nature Communications.

Reviewer Comments:

Reviewer #1 (Remarks to the Author):

The article by Stashko et al. investigates the possibility of developing an automatized atomic force microscopy (AFM) technique with the intention to obtain reliable spatial information about tissue stiffness. Briefly, they created a motorised stage for AFM, worked with PI-stained samples (to guide imaging and stitch images) and measuring PA gels of defined stiffness before and after each imaging (for calibration). They further overlaid these images with high magnification DAPI stain, used a fluorescent collagen binding protein to stain collagen structures and developed a convolutional neural network expose tissue elasticity onto collagen morphological features. After validating their system, the authors investigated stiffness associated metastatic potential using human HER2+ PDX samples and further HER2+ or TNBC primary breast cancer samples. Their results suggest the new technique (STIFmap) can predict metastatic potential. Additionally, STIFmaps allowed spatial association of ECM stiffness to EMT gene expression. This is a well thought and well invested study to make AFM an easy and mainstream technique to measure tissue stiffness. This new technique allows identification of mechanical hotspots where cells interact with collagen rich ECM, therefore will add to spatial gene expression knowledge base in a contextual manner. The article is well written and presented. Figures were informative.

Comments:

1. Theoretically, HER2+ tumours should have active Ras pathway and therefore have abundant P-ERK. Is there another and more specific way to assess the effects of ECM stiffness and integrin signalling in PDX modelling? For example, survivin and bcl-2 upregulation or p27 downregulation were shown to be positively regulated by integrin signalling. In addition, YAP/TAZ signalling can be investigated in this context.

We agree that HER2+ tumors exhibit activated Ras signaling and elevated phospho-ERK. However, we and others showed that a stiff ECM further increases phospho-ERK activity by enhancing growth factor receptor signaling. Nevertheless, to address the reviewers concerns, in our revised manuscript we incorporated additional markers indicating elevated mechanosignaling including staining for and quantification of nuclear YAP localization. These new data are included in a revised figure 4.

2. The use of STIFMaps in FFPE is investigated but data provided is limited. I understand morphological features of cryopreserved or FFPE tissues were reflected through CNA collagen staining but to address the problem correctly, tissue stiffness and collagen structure should be assessed before and after FFPE staining and assessing a correlation in a true experimental model.

The reviewer raises an interesting point. We would like to respectfully point out that a pathologist evaluated collagen morphology in a patient-matched cryopreserved and FFPE sample and did not

discern any morphological or pathological differences (Fig S5). Additionally, in our study we utilized pathology-archived FFPE tissues, thus it is impossible to conduct staining within the same tissue before and after formalin-fixation. However, in an effort to address the reviewer's comment, we did image collagen with CNA-35 staining in cryopreserved tissues before and after 10% formalin fixation for 1 hour. Importantly, we did not observe any significant morphological differences following image overlay (red; pre-fixation and blue; post-fixation; see images below).

We further confirmed that antigen retrieval (AR) does not disrupt collagen morphology. We stained collagen with Picrosirius Red staining in FFPE tissues before and after antigen retrieval (red; pre-AR and blue; post-AR; see figure inserted below). We incorporated these images into our revised manuscript, as extended data figure 5 (panel b and c) and discussed the results in the main text.

3. Are there any morphological or histopathological differences in mets of PDX tumours embedded in soft or stiff material?

While this question is intriguing we respectfully contend that this question falls well beyond the scope of the current manuscript. Indeed, we already presented detailed data quantifying the frequency of mice with lung metastasis, and reported on the number and size of metastatic lesions in the HER2+ PDX tumors. The presence of metastasis is clinically an independent poor prognostic indicator. Moreover, the morphology of lung metastasis has no bearing on clinical outcome. Nevertheless, to address the current reviewers concerns, in our revised manuscript we incorporated histopathological assessment of the primary HER2+ PDX tumor samples into the main text. Our histopathological analysis identified all HER2+ PDX tumors to be of high histological grade. However and importantly, we did note that there was more extensive necrosis in the STIFF ECM tumors, which further supports that stromal stiffness drives tumor aggression. Finally, given metastatic lesions are thought to undergo mesenchymal-to-epithelial transition, investigating EMT markers histologically may yield non interpretable results that would require extensive studies to address and would go well beyond the scope of the current manuscript (reviewed in doi.org/10.1084/jem.20181827).

4. The authors stained TNBC tissues with ZEB1 and SLUG, quoting an article in melanoma. Breast cancer and melanoma are very different in nature. Nuclear localisation of Twist due upon exposure to stiffer ECM has been reported in breast cancer and accepted as a reason of stiffness associated EMT (Wei et al 2015, Fattet et al. 2020). This aspect of Twist1 (nuclear localisation) cannot be assessed by RNA abundance. It is a good idea to investigate Twist1 expression (rather than SLUG), especially at tumour-stroma interface, along with STIFmap technique in patient materials.

We now include additional references describing EMT in breast cancer. We acknowledge that RNA abundance does not necessarily reflect protein abundance nor can it assess protein nuclear localization. To address this concern in our revised manuscript we performed Twist1 staining and combined this with our STIFMap method. The new data clearly demonstrates that elevated total and nuclear Twist1 associates with stromal stiffness in TNBC tissues ($****P < 10^{-5}$). We respectfully contend that the elevated expression we previously quantified using RT-PCR and reported in our original submitted manuscript does in fact support our finding that STIFF ECM tumors correlate with markers of a mesenchymal phenotype.

5. Can STIFmap technique determine collagen bundling? There are plenty of reports suggesting collagen bundling can affect or determine metastatic potential.

The reviewer raises a very interesting question. Identifying collagen bundling with STIFMap still requires further work, but we do believe that the model can be trained to predict collagen bundling. Indeed, we are currently considering this option, however, to decouple collagen stiffness and collagen bundling is not trivial. A recent article (<https://doi.org/10.1038/s41388-022-02258-1>) using organotypic models from mice, has been able to decouple bundling and stiffness in vivo, which would provide an elegant way to validate the trained model. We are exploring the use of in vitro systems for collagen bundling to test it experimentally (E.g. <https://doi.org/10.1021/acsomega.9b03704>).

6. Please italicize Latin phrases (in situ, etc)

In our revised manuscript we italicized all latin phrases.

Reviewer #2 (Remarks to the Author):

In this study, the authors estimate tissue stiffness with a microscopic technique. To the reader it is unclear if the main advance is the biological understanding or the method per se. The whole article is a bit confusing at the first read. Only after reading the whole thing the reader realizes that this is actually

mostly an engineering article which reports a new AFM protocol with some data analysis. To me it is unclear who the readership of this article would be, apart from people in the AFM field.

Comments:

1. The phrasing of the title is odd. The message should focus on the biological finding, and could mention the tool used to make this finding. Please rephrase.

We retain that the title our of manuscript is clear and concise, highlighting the utility of our STIFMap tool in identifying mechanical heterogeneity spatially within tissues. We thereafter used our STIFMap tool to reveal a direct association between stromal stiffness and EMT in human breast tumors.

2. In the abstract, abbreviations such as AFM are not defined. The abstract is confusing to read. It is again unclear what was the state of the art before the study and what's the novelty and the key finding. The last sentence of the abstract does not seem to be substantiated by data.

We revised our manuscript text and carefully defined all abbreviations used within the abstract and throughout the main text. Within our revised abstract, we highlighted the utility of STIFMaps to enable spatially resolved forced maps across whole tissue sections which we were able to integrate with biomarker staining to identify the novel association between stromal stiffness and EMT, which has not been shown before. In our revised introduction, we now restrict our discussion of previous methods towards work done by prior investigators to quantify tissue rheology.

3. The introduction is a bit excessive, trying to link many biological concepts, but in my opinion it should focus more on the topic at hand, the mechanical characterization of tissue.

We maintain that our introduction provides a cohesive summary regarding tumor heterogeneity, stromal stiffness and EMT in driving breast cancer aggression and poor patient outcome. We also provide a background of current state of the art methods to study spatial tumor heterogeneity and tissue rheology. We include this information to prime a broad range of readers to understand the methodology and experimental findings presented within our manuscript.

4. The results report the outcome of a lot of experimental work, which is not directly related to the topics discussed in the introduction. The results have too much technical details which should be moved to the methods section.

We provide only essential technical detail within the results so that we can provide the reader with an understanding of all of the data presented in the results.

5. The figures have a high visual quality.

We thank the reviewer for this comment.

Reviewer #3 (Remarks to the Author):

This manuscript proposes an automated AFM system capable of measuring the mechanical distribution of breast cancer tissue slides which combined with other traditional biological and clinical detection and analysis methods, to study the relationship between tissue stiffness heterogeneity and breast cancer tumor aggression behavior. The AFM mechanical measurement data, corresponding CNA and DAPI images are used to train the CNN network to generate the elasticity distribution of tissue sections. The proposed STIFMaps method also has the potential to predict the mechanical distribution of formalin-Fixed paraffin-embedded tissue sections that have lost their intrinsic mechanical information.

Experiments show that the STIFMaps obtained by the CNN network is consistent with the information extracted by biological methods and can reveal the relationship between the stromal stiffness of human breast tumor specimens and human breast cancer aggression. This is a very interesting paper and provides a promising way to expand AFM mechanical measurement from cell-scale to tissue-scale. In addition, the STIFMaps solves the problem that traditional AFM cannot perform mechanical mapping of clinical tissue specimens and makes a great contribution to the field of AFM. At the same time, it also provides a versatile tool for clinical to analysis, treat and predict breast cancer. Here are some comments.

1. The accuracy of AFM measurements of mechanical properties of tissue section samples should be considered. The article discussed the case where the cantilever was contaminated. But the tip which is the 5 μ m radius sphere is more easily contaminated and contributes more to the potential bias in the modulus measurement result. The contact area and the tip modulus have a major influence on the modulus calculated from the contact model such as Hertz model. If the surface of the sphere is contaminated, the shape and surface area will change. In addition, if the modulus gap between the sphere and the contamination is large, the bias of measurement result should be considered and discussed. It is recommended that the author characterize the microspheres after multiple measurements or perform secondary calibration on the standard sample to eliminate the influence of tip contamination on measurement results.

We agree with the reviewer that the tip can become contaminated. Accordingly, to assess the integrity and consistency of the tip in our studies, we utilized polyacrylamide gels across a range of known (and measured) stiffnesses as reference samples. We evaluated the stiffness of the polyacrylamide gels both before and after each day of AFM experimental measurement. By comparing the stiffnesses pre- and post-sampling, we observed a 1:1 relationship (see attached figure), indicating no changes in stiffness were observed. If the tip was contaminated, we would expect the data to be consistently above/below the 1:1 line (140 Pa gels p-value = 0.551 and 1 kPa gels p-value = 0.970. Wilcoxon Rank-Sum Test. Normality was checked using Shapiro-Wilks Normality Test). Therefore, we conclude that the tip was not contaminated in our studies, and that the data shown in our manuscript is not affected by tip contamination. We have incorporated this figure into the revised manuscript, in an extended data figure 2 and discussed these controls in the main text.

2. The surface of the thawed tissue slides sample will not be as smooth as that of the single cell sample. If the fluctuation of the sample surface topography is larger than 5 microns which is the diameter of the sphere used in this article, it is likely that the cantilever beam will touch the sample surface first rather than the sphere. In this case, the obtained force curve cannot be applied to the contact model based on sphere to calculate the sample modulus. Therefore, it is necessary to

characterize the surface of the sliced tissue to evaluate the validity of the force curves acquired by the sphere of 5um diameters. Maybe the diameter of the sphere should be increased or the flatness of the tissue sample surface should be further improved.

We appreciate the reviewer's comment, however, we do not believe that the beam is touching the sample prior to the sphere. To support this assertion, we maintain that the cantilevers are tilted between 10-20 degrees (<https://doi.org/10.1021/la036128m>). We measured the cantilevers used for the measurement we reported in this work, whereby the distance between the sphere and the beam is around 45 um (see attached figure, scale bar; 50 um).

In the worst case scenario, when the cantilever is tilted 10 degrees, we can still obtain the clearance of the z-axis, which would result in $C = \sin(10) * 45 = 7.8 \mu\text{m}$, therefore providing even a higher distance that avoids touching with the beam. We experimentally tested this assumption. To do so, we did AFM measurements in a PRIMO PDMS system, indenting with the sphere and with the beam. We found that the sphere is 5.5 um lower than the tip of the cantilever beam, in agreement with our analytical solution. We don't expect a local variation in height of 5.5 um (27.5% of the section height)

Furthermore, when using the Hertz model for the calculation of the stiffness, the stiffness is given by the following equation:

$$E = \frac{k \Delta d^3 (1 - \nu^2)}{4 R^{1/2} \delta^{3/2}}$$

Where k is the spring constant of the cantilever, Δd is the cantilever deflection, ν is the Poisson ratio of the sample, R is the radius of the tip and δ is the indentation. Therefore, the stiffness is inversely proportional to the radius of the tip (R) and directly proportional to the ratio of $\Delta d/\delta^{3/2}$.

If the sample is touched with the beam, we can assume that the radius of the tip is either at least one order of magnitude higher (width of the beam $\sim 30 \mu\text{m}$) or tends to infinity (<https://doi.org/10.1038/ncomms11566>), therefore the stiffness of the tissue would significantly drop, or to keep reasonable values for the stiffness, the ratio of $\Delta d/\delta^{3/2}$ would significantly increase. We did not observe any of these situations in our measurements.

3. The thickness of the specimens used were $20 \mu\text{m}$. The indented depth was $2 \mu\text{m}$ for each force curve. The substrate effect will affect the accuracy of the Modulus result if the indentation depth is too large. The suitable value of indentation depth is considered as less than 10% of the sample thickness. In this paper, the indentation-thickness ration is 10%, so I suggested that the substrate effect should be considered and studied by decreasing the indentation depth. Besides, the typical force curves in various positions of the specimen should be provided in supplementary information or extended data. The difference in the obtained mechanical properties results can be evaluated from the force curves.

We would like to highlight that all the curves are available in <https://data.mendeley.com/datasets/vw2bb5jy99/2>, we clarified that in the revised manuscript (DATA AVAILABILITY).

We respectfully point out that according to Buckle's rule, **the indentation should not exceed 10%, which is the criteria we used for our measurements**. Nevertheless, to rigorously address reviewer's comment, we reassessed all of our AFM curves and found that for 95.4% of the samples, the indentation was less than 10%.

For the remaining 4.6% of the data, we reanalyzed the data fitting 2 microns to adhere to Buckle's rule. This analysis yielded virtually the same results as the existing stiffnesses (Pearson's $r = 0.995$).

Finally, we compared all the data limiting the maximum indentation to 1 μm after the contact point vs 2 μm after the contact point (5% and 10% sample thickness, respectively). Once again we saw that the choice of maximum indentation depth made little difference on the resulting stiffness measurements (Pearson's $r = 0.998$). Altogether, these data indicate that our results are not affected by the substrate effect. We have included this information the Methods (Atomic Force Microscopy)

- Using STIFMaps to predict the tissue mechanical distribution of Formalin-Fixed Paraffin-Embedded Tissue is one of the contributions in this paper, which will also provide potential contribution to the pathology and clinical medicine. But the authors didn't provide further experimental results to prove the feasibility of this method. It is suggested that the comparison of the modulus between the measured fresh sample and the predicted FFPE sample should be added. This will significantly improve the quality of the article.

As reported in our manuscript the utility of STIFMaps was used to assess mechanical heterogeneity across whole-slide images of human FFPE clinical specimens. We were then able to link stiffness measurements to register with EMT marker analysis that supported a significant correlation between regions of elevated ECM stiffness and indicators of an epithelial to mesenchymal transition phenotype. While it would be interesting to understand how formalin-fixation alters the mechanical properties of the tissue, we are unable to compare the stiffness measurements of these samples pre- and post-fixation since we utilize pathology-archived FFPE tissues. Instead, we carefully addressed the modifications we incorporated into our revised manuscript similar to our response to reviewer ones comments on this same topic – see reviewer #1 comment #2 and associated rebuttal.

- For the CNN used in this paper, it is mentioned that Alexnet has a better effect than other networks. However, the performance of neural network ResNet and DenseNet in images classification is better than that of AlexNet proposed in 2012. So my question is why AlexNet has better effect than other networks on your task? Regarding the use of AlexNet in the pytorch library, whether a pre-trained model is used (the parameters trained by AlexNet on a large data set are loaded during initialization), and how many images are used during training, and whether there is any overfitting during the training process? (The performance on the train set is much better than the test set).

While it is true that DenseNet and ResNet perform better on image classification tasks, to our knowledge they have not been applied to regression tasks and therefore it is unknown if they perform better at this style of problem than AlexNet. The AlexNet utilized in these studies is a simpler model with fewer parameters than DenseNet or ResNet, making it more straightforward to tune hyperparameters and visualize intermediate layer activations. Moreover, the added complexities of DenseNet and ResNet also entail longer training times and greater computational power, making

AlexNet faster to optimize. Nevertheless, we did experiment with DenseNet and ResNet architectures, though these models each took several days to train, preventing us from optimizing further:

The applied AlexNet was not pre-trained. Weights and biases are initialized using Xavier initialization, which is the PyTorch default. The model performs better on the validation set than the training set (Fig. 2c) due to the transformation that are applied to the training set to avoid overfitting (Fig. 2a). All data is available via <https://data.mendeley.com/datasets/vw2bb5jy99/2> (refer to DATA AVAILABILITY STATEMENT in manuscript) for training, validating, and reproducing the exact model architecture used in these studies for users interested in experimenting with different models such as DenseNet and ResNet.

REVIEWER COMMENTS

Reviewer #1 (Remarks to the Author):

The article by Stashko et al. has been mostly revised according to my recommendations. However, I should say that I am disappointed with the revised manuscript for 2 reasons.

1- The authors provided very little or no guidance as to how the manuscript was revised or where in text the revision has taken place. In the rebuttal letter they were using vague language to suggest they made changes but it is impossible to identify what was done. For example they say "in our revised manuscript we incorporated additional markers indicating elevated mechanosignaling including staining for and quantification of nuclear YAP localization". I have no guidance which additional markers were added and in which panels?

Overall, the changes were not highlighted, there was no guidance to identify the revision in text and figures. I trust the authors have clarified certain issues in the revised version of the manuscript but the whole meaning of review process goes away if I cannot find what I am looking for.

2- Two referees, (me and ref 3) highlighted the fact that the new technique has not been empirically tested in FFPE tissues (my comment 2 and referee 3 comment 4) although the authors specifically claim their approach will "advance current stiffness measurement techniques which preclude FFPE samples". The novelty aspect of the paper is the possibility that one can measure the stiffness of an FFPE fixed tissue and identify its stiffness before fixation. This will allow a clinical use for AFM and retrospective studies.

I believe there has been a misunderstanding between the authors and me (and referee 3). To address our concerns, the authors compared the ECM morphology of FFPE tissue with ECM staining and AFM imaging as analysed by a pathologist. We all know that FFPE preserves the structural identity of the tumour so I do not dispute their findings on morphology front. But there is no data in the manuscript about tissue stiffness before and after fixation. Therefore their answer is not addressing my question or justifying their claim. This question could only be empirically addressed by analysing the same tissue, before and after formalin fixation, using conventional pathology and with AFM. I believe duration of fixation and age of the FFPE block are also variables in this equation which should be tested. I would expect formalin fixation would increase the stiffness of the tissue as it is a cross linking agent. In that regard maybe a correlation (e.g. fold increase) can be found between post and pre fixation stiffness.

Reviewer #2 (Remarks to the Author):

The authors did not make substantial changes in response to my comments. They just argued that they do not agree with my comments. It is OK to have different opinions on scientific questions. However, I maintain that the article would benefit from the clarifications I suggested.

Reviewer #3 (Remarks to the Author):

The reviewer's comments were seriously considered by the authors. The paper was revised accordingly. The contributions of the paper is clear and the paper is well written and organized.

We thank the reviewers for their re-assessment of our revised manuscript. We have responded to each of the newly raised issues in a point by point rebuttal listed below. We are confident the reviewers will be satisfied with our updated manuscript revisions and will agree that our article is now suitable for publication in Nature Communications.

Reviewer Comments:

Reviewer #1 (Remarks to the Author):

The article by Stashko et al. has been mostly revised according to my recommendations. However, I should say that I am disappointed with the revised manuscript for 2 reasons.

1. The authors provided very little or no guidance as to how the manuscript was revised or where in text the revision has taken place. In the rebuttal letter they were using vague language to suggest they made changes but it is impossible to identify what was done. For example they say “in our revised manuscript we incorporated additional markers indicating elevated mechanosignaling including staining for and quantification of nuclear YAP localization”. I have no guidance which additional markers were added and in which panels? Overall, the changes were not highlighted, there was no guidance to identify the revision in text and figures. I trust the authors have clarified certain issues in the revised version of the manuscript but the whole meaning of review process goes away if I cannot find what I am looking for.

We apologize for not providing a version of our revised manuscript text file with tracked changes. We have now color highlighted all of the changes that were made in the text of our revised manuscript. Within our original point-by-point response to each of the reviewer’s comments, where possible we referenced the Figure that included modifications to the data. With respect to reviewer #1s original comments, in our newly revised manuscript as requested we now list where significant modifications occur to the Figures and/or lines of text in response to major comments:

Comment 1: These new data are included in revised Figure 4, and text modifications occur within the results section on lines 307-310 and materials and methods section on lines 651-652.

Comment 2: These new data are included in revised Extended Data Figure 5, and text modifications occur within the results section on lines 284-291.

Comment 3: Text modifications occur within the results section on lines 314-316.

Comment 4: These new data are included in revised Figure 5, and text modifications occur within the results section on lines 337-339 and materials and methods section on lines 653-654.

2. Two referees, (me and ref 3) highlighted the fact that the new technique has not been empirically tested in FFPE tissues (my comment 2 and referee 3 comment 4) although

the authors specifically claim their approach will "advance current stiffness measurement techniques which preclude FFPE samples". The novelty aspect of the paper is the possibility that one can measure the stiffness of an FFPE fixed tissue and identify its stiffness before fixation. This will allow a clinical use for AFM and retrospective studies. I believe there has been a misunderstanding between the authors and me (and referee 3). To address our concerns, the authors compared the ECM morphology of FFPE tissue with ECM staining and AFM imaging as analysed by a pathologist. We all know that FFPE preserves the structural identity of the tumour so I do not dispute their findings on morphology front. But there is no data in the manuscript about tissue stiffness before and after fixation. Therefore, their answer is not addressing my question or justifying their claim. This question could only be empirically addressed by analysing the same tissue, before and after formalin fixation, using conventional pathology and with AFM. I believe duration of fixation and age of the FFPE block are also variables in this equation which should be tested. I would expect formalin fixation would increase the stiffness of the tissue as it is a cross linking agent. In that regard maybe a correlation (e.g. fold increase) can be found between post and pre fixation stiffness.

Within the original reviewer comments, this reviewer requested that we "assess tissue stiffness and collagen structure before and after formalin-fixation paraffin-embedding (FFPE)". We agree with the reviewer that FFPE processing preserves collagen morphology, which we demonstrate in Extended Data Figure S5 through pathological assessment of collagen morphology between patient-matched cryopreserved and FFPE samples, as well as pre- and post-formalin-fixation in cryopreserved samples. However, with all due respect, in our newly revised manuscript, as requested by this reviewer, we did not assess tissue stiffness pre- and post-formalin-fixation. May we respectfully point out that formalin fixation crosslinks the tissue thereby dramatically increasing the stiffness of the tissue to such a degree that it would be near impossible to distinguish differential regions of stiffness versus compliance in any given tissue. For instance, it has been shown that physiological differences in the stiffness of fresh tissue are lost upon formalin-fixation (Calò, A., Romin, Y., Srouji, R. *et al.* Spatial mapping of the collagen distribution in human and mouse tissues by force volume atomic force microscopy. *Sci Rep* **10**, 15664 (2020). <https://doi.org/10.1038/s41598-020-72564-9>, Fig. 2). Moreover, as requested by the current reviewer, we also respectfully point out that post-formalin fixation stiffness depends upon many variables: tissue content, tissue age, fixation method, fixation duration, among other factors. These are all highly variable across various clinical pathology settings. This has led us to conclude that executing the measurements as requested by the current reviewer would fail to address their main concerns and moreover, would not add any physiological relevance to the findings of our study. We suggest that this very issue regarding FFPE processing actually highlights the utility of our STIFMap method, which preserves the integrity of the collagen morphology, and thereby permits the prediction of the mechanical properties of these clinical tissues. Indeed, the very rationale for developing the STIFMaps method is to overcome these stated technical issues and to permit analysis of FFPE clinical tissues including assessing the mechanical heterogeneity of the specimen across whole-slide images of FFPE clinical specimens.

The current correctly pointed out that reviewer 3 raised similar points pertaining to the fidelity of the collagen architecture and our AFM measurements. Accordingly, we addressed the similar issue raised by reviewer 3 and include our rebuttal to reviewer 3 for this reviewer's assessment. and their acknowledgement that we comprehensively addressed the points they raised.

“As reported in our manuscript the utility of STIFMaps was used to assess mechanical heterogeneity across whole-slide images of human FFPE clinical specimens. We were then able to link stiffness measurements to register with EMT marker analysis that supported a significant correlation between regions of elevated ECM stiffness and indicators of an epithelial to mesenchymal transition phenotype. While it would be interesting to understand how formalin-fixation alters the mechanical properties of the tissue, we are unable to compare the stiffness measurements of these samples pre- and post-fixation since we utilize pathology-archived FFPE tissues. Instead, we carefully addressed the modifications we incorporated into our revised manuscript similar to our response to reviewer ones comments on this same topic – see reviewer #1 comment #2 and associated rebuttal.”

We respectfully point out that reviewer 3 agreed and acknowledged our rebuttal to their query and stated that we had comprehensively addressed the points they had raised including the fidelity of the collagen architecture and our AFM measurements as similarly raised by this current reviewer.

Reviewer #2 (Remarks to the Author):

The authors did not make substantial changes in response to my comments. They just argued that they do not agree with my comments. It is OK to have different opinions on scientific questions. However, I maintain that the article would benefit from the clarifications I suggested.

We are in the process of carefully editing the manuscript to clarify the “general” points raised by reviewer 2.

Reviewer #3 (Remarks to the Author):

The reviewer's comments were seriously considered by the authors. The paper was revised accordingly. The contributions of the paper are clear and the paper is well written and organized.